# A melanopsin ganglion cell subtype forms a dorsal retinal mosaic projecting to the supraoptic nucleus

Michael H. Berry [1,2], Michael Moldavan [3,4], Tavita Garrett [1,5], Marc Meadows [5,6], Olga Cravetchi[3,4], Elizabeth White[1], Joseph Leffler[1], Henrique von Gersdorff [1,6,7], Kevin M. Wright [6], Charles N. Allen [3,4] & Benjamin Sivyer [1,7] ✉

Visual input to the hypothalamus from intrinsically photosensitive retinal ganglion cells (ipRGCs) influences several functions including circadian entrainment, body temperature, and sleep. ipRGCs also project to nuclei such as the supraoptic nucleus (SON), which is involved in systemic fluid homeostasis, maternal behavior, social behaviors, and appetite. However, little is known about the SON-projecting ipRGCs or their relationship to well-characterized ipRGC subtypes. Using a *GlyT2^{Cre}* mouse line, we show a subtype of ipRGCs restricted to the dorsal retina that selectively projects to the SON. These ipRGCs tile a dorsal region of the retina, forming a substrate for encoding ground luminance. Optogenetic activation of their axons demonstrates they release the neurotransmitter glutamate in multiple regions, including the suprachiasmatic nucleus (SCN) and SON. Our results challenge the idea that ipRGC dendrites overlap to optimize photon capture and suggests non-image forming vision operates to sample local regions of the visual field to influence diverse behaviors.

In addition to the rod and cone photoreceptors that are used for image-forming vision, the mammalian retina contains intrinsically photosensitive retinal ganglion cells (ipRGCs) that primarily drive non-image-forming behaviors[1,2]. ipRGCs express their own photopigment, melanopsin[3], and project to a diverse array of central brain regions[4–6], influencing many homeostatic functions, including circadian entrainment, pupil constriction, body temperature, sleep, and mood[7–11]. There are six main types of ipRGCs (M1–M6), which are categorized according to their dendritic morphology, melanopsin expression, gene expression, and central projection locations[6,12]. The most studied of these, the M1 ipRGCs, have dendrites that primarily occupy sublamina-a of the inner plexiform layer (IPL). They form the primary projections to the suprachiasmatic nucleus (SCN), which is the master circadian clock[4,7,13–15], and the shell of the olivary pretectal nucleus (OPN), which serves as the primary site of light-dependent pupillary constriction[8,16,17]. They also project to a number of lateral hypothalamic brain regions, such as the supraoptic nucleus (SON), ventral lateral preoptic area (VLPO), and medial amygdaloid nucleus, though the functional role of these projections remains unclear[4,6,15].

The retinal responses of M1 ipRGCs are suited to their primary role in non-image-forming vision; their long and sustained responses to bright illumination reflect their comparatively high expression of melanopsin and weak photoreceptor-mediated synaptic drive from retinal bipolar cells[6,18]. These light responses are optimal for signaling

[1]Department of Ophthalmology, Casey Eye Institute, Oregon Health & Science University, Portland, OR, USA. [2]Medical Scientist Training Program, Oregon Health & Science University, Portland, OR, USA. [3]Oregon Institute of Occupational Health Sciences, Oregon Health and Science University, Portland, OR, USA. [4]Department of Behavioral Neuroscience, Oregon Health & Science University, Portland, OR, USA. [5]Neuroscience Graduate program, Oregon Health & Science University, Portland, OR, USA. [6]Vollum Institute, Oregon Health & Science University, Portland, OR, USA. [7]Department of Chemical Physiology and Biochemistry, Oregon Health & Science University, Portland, OR, USA. ✉e-mail: sivyer@ohsu.edu

absolute light intensity and driving behaviors that are slow, such as circadian entrainment, and the maintained component of the pupillary light reflex[16,19,20]. M1 ipRGCs were first thought to comprise a single, homogenous population. However, the discovery of a subpopulation lacking *Brn3b* expression[17], the divergent projection patterns of M1 ipRGCs according to *Brn3b* expression[15,17], and the diversity of light responses within M1 ipRGCs[21,22] together suggest there are multiple M1 subtypes mediating different roles in non-image-forming behavior.

The high density of M1 ipRGCs in the retina also suggests they comprise multiple subtypes. Conventional RGCs within a functional subtype are commonly arranged in evenly spaced mosaics where their dendrites form territories with minimal overlap[23,24]. This arrangement is thought to optimize the sampling of visual space[25-27] and reduces the encoding of redundant information, where each RGC subtype samples an even component of the visual field across the retina. Previous reports indicate M1 ipRGC dendrites are not territorial, and they overlap considerably—about fourfold[28]. This might be due to their non-image-forming role, where the even representation of visual space is forgone in favor of increasing their dendritic surface area, thus maximizing the surface area for photon capture. Alternatively, they might comprise multiple functional subtypes, each of which independently tiles the retina. We provide evidence for the latter, illustrating that, like conventional RGCs, ipRGCs are arranged in mosaics optimal for the even representation of visual space. However, the retinal distribution of ipRGCs and how this relates to specific subtypes of M1 ipRGCs has remained elusive.

Here we provide evidence that, like conventional RGCs, ipRGCs are arranged in a tiled mosaic optimal for the even representation of visual space. We use a combination of mouse genetics, confocal microscopy, anterograde and retrograde labeling, patch-clamp recordings, and optogenetics to describe a subtype of M1 ipRGCs that are found only in the peripheral dorsal retina. They form a territorial mosaic within this region, suggesting mice devote additional melanopsin-dependent processing power to their ventral visual field. This subtype of M1 ipRGCs forms the primary visual projection to the SON, and projects to unique sub-regions of other non-image-forming brain nuclei, like the SCN and IGL, where they release the excitatory neurotransmitter glutamate, despite having *Cre* expression driven by a BAC transgene containing the promotor of the glycine transporter GlyT2.

## Results

### A population of ipRGCs encoding ventral vision

We discovered RGCs in mice where *Cre* is driven by a BAC encoding the inhibitory glycine transporter GlyT2 (slc6a5 KF109; ref. [29] Fig. 1a). In these mice, *Cre* is expressed both in GABAergic and glycinergic neurons in the retina and brain[30]. In the retina, *Cre* expression is overwhelmingly restricted to inhibitory amacrine cells such as the glycinergic AII amacrine cell[31]. However, we also observed fluorescent axons in the ganglion cell layer (Fig. 1a), and when following them to the optic nerve head, discovered they originated solely from RGCs in the dorsal retina (Fig. 1c). Hypothesizing that these RGCs likely represent a unique population in the retina, we sought to determine their functional identity by mapping their central axonal projections in the brain with *Cre*-dependent anterograde labeling of their axon terminals and their light responses and dendritic morphology with targeted electrophysiological recordings and Neurobiotin fills.

To determine the location of the central projections of their axon terminals, we injected an AAV into the eye, enabling the *Cre*-dependent expression of fluorescent protein (Fig. 1b). The axons of RGCs labeled using this method predominantly innervated non-image-forming brain regions such as the intergeniculate leaflet (IGL) and suprachiasmatic nucleus (SCN) (Fig. 1d), suggesting they arose from ipRGCs, which form the predominant projections to these regions. To confirm their identity in the retina, we performed melanopsin antibody co-staining in *GlyT2Cre;Ai140* mice (Fig. 1e) and *GlyT2Cre;Ai9* mice (Fig. S1) and found that fluorescent RGCs in the dorsal retina co-expressed melanopsin. We subsequently targeted fluorescent cell bodies in isolated preparations of the dorsal retina from *GlyT2Cre;Ai9* mice for electrophysiological action potential recordings. Current-clamp recordings from fluorescent somas allowed us to confirm intrinsically photosensitive action potential responses in the presence of a cocktail of excitatory synaptic blockers (Fig. 1f; 20 μM L-AP4, 25 μM DAP5, and 20 μM CNQX).

To identify the dendritic morphology of these ipRGCs, we performed cell-targeted Neurobiotin fills from *GlyT2Cre;Ai9* retinae. These experiments revealed they are predominantly comprised of ipRGCs stratifying in sublamina-a of the IPL, a feature unique to M1 ipRGCs[32,33] (Fig. 2a). We also found they contained a second population stratifying in the sublamina-b of the IPL, with variable morphologies,

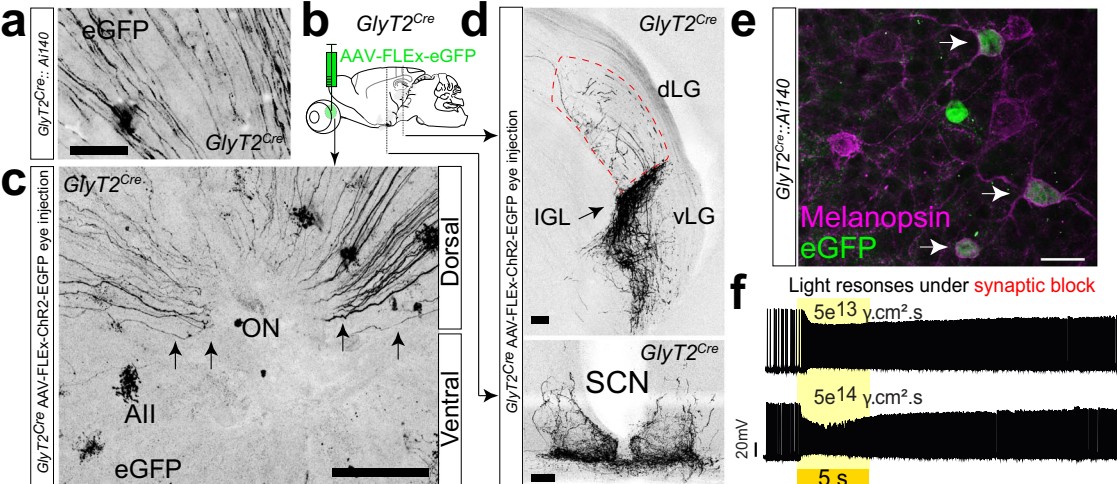

**Fig. 1 | A subpopulation of ipRGCs in the dorsal retina. a** RGC axons identified in the dorsal portion of *GlyT2Cre;Ai140* whole-mount retina. **b, c** Cre-dependent virus injection into the eyes of *GlyT2Cre* mice labels RGC axons that project from the dorsal retina via the optic nerve (ON) to non-image-forming central areas. **d** Intergeniculate leaflet (IGL; top) and suprachiasmatic nucleus (SCN; bottom; dLGN and vLGN dorsal and ventral lateral geniculate nucleus. **e** Confocal micrographs of melanopsin antibody staining in *GlyT2Cre;Ai140* mice labeling cells with EGFP. **f** Current-clamp recordings of light responses to 5 s visual stimuli under the synaptic block (20 μM L-AP4, 25 μM DAP5, 20 μM CNQX) illustrate *GlyT2Cre*-positive RGCs are intrinsically photosensitive (ipRGCs). Similar results were obtained from eight recordings in three mice. Scale bar in **a** = 20 μm, **c** = 100 μm, **d** = 100 μm, **e** = 20 μm. Source data for panel **f** are provided as a source data file.

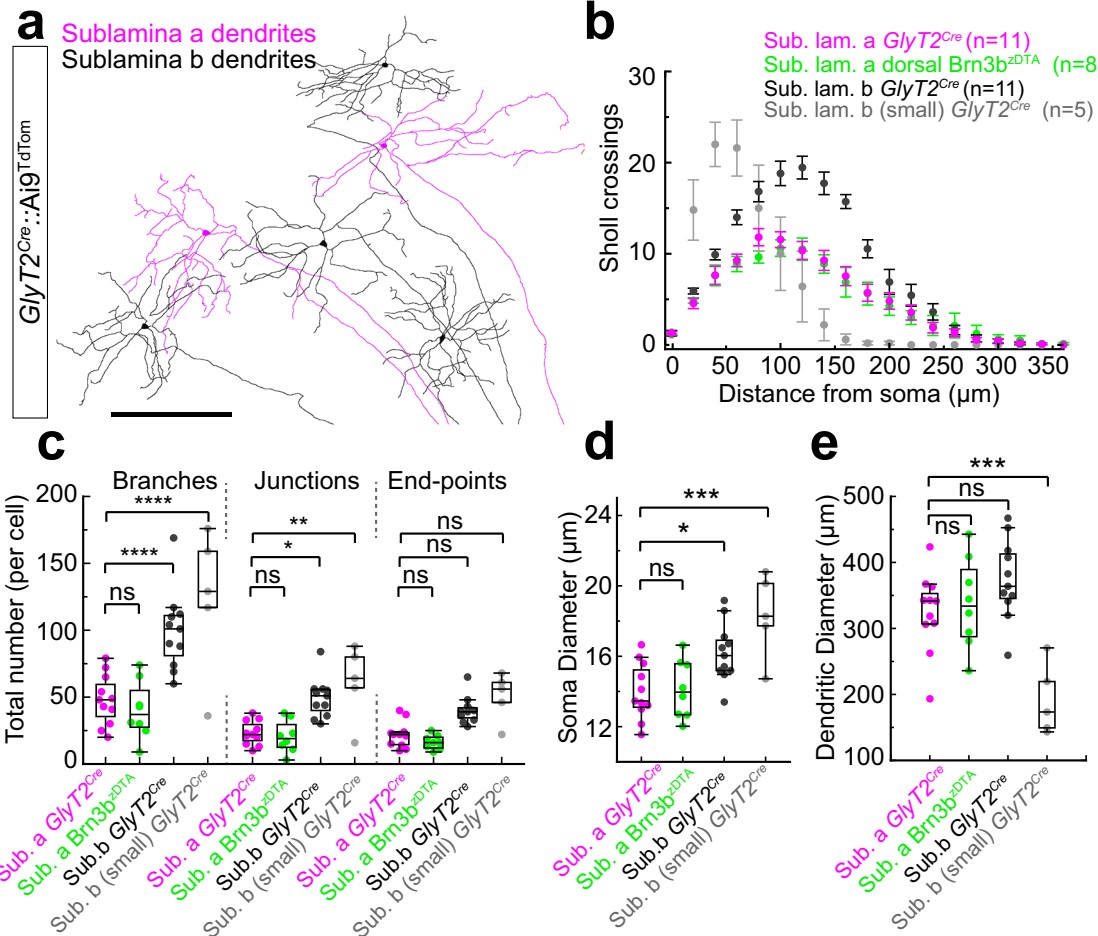

**Fig. 2 | Dendritic morphology of *GlyT2^Cre* ipRGCs. a** Tracings of Neurobiotin electroporated *GlyT2^Cre* ipRGCs illustrate multiple morphological subtypes with dendritic stratification in sublamina-a or sublamina-b of the inner plexiform layer (IPL). **b** Dendritic crossings at radial distances from each soma (Sholl analysis). **c** Number of branches, junctions and end-points (top $p < 0.0001$; middle $p < 0.001$; ns $p = 0.9928$; one-way ANOVA Šídák's multiple comparisons test). **d** Soma diameter (***$p = 0.0002$; **$p = 0.0175$; ns $p = 0.997$; one-way ANOVA Šídák's multiple comparisons test). **e** Dendritic diameter (***$p = 0.001$; ns middle $p = 0.1866$; ns bottom $p = 0.9573$; one-way ANOVA Šídák's multiple comparisons test).

*GlyT2^Cre* sublamina-a stratifying ipRGCs (M1; $n = 11$ cells from $n = 5$ animals), *OPN4^Cre;Brn3b^zDTA* ipRGCs (M1; $n = 8$ cells from $n = 1$ animal), *GlyT2^Cre*-positive sublamina-b stratifying ipRGCs (M2; $n = 11$ cells from $n = 5$ animals), small sublamina-b stratifying ipRGCs (M4/5; $n = 5$ cells from $n = 4$ animals). Dendrites of *OPN4^Cre;Brn3b^zDTA* M1 ipRGCs were reconstructed from the antibody-stained retina. Dendrites of M1, M2, and some M5 ipRGCs were reconstructed from Neurobiotin-filled cells in *GlyT2^Cre;Ai9* mice. Data were presented as mean values ± SEM. Box and whisker parameters are provided in Supplementary Table 3 and in a source data file. Scale bar in (**a**) = 300 μm. Source data are provided as a source data file.

resembling a mixture of non-M1 ipRGCs[34–37]. Characterizing their dendritic structure using Sholl analysis (Fig. 2b), we found that the morphological complexity of the sublamina-a stratifying cells, including the total number of branching points, junctions, and end-points are distinct from the mixture of sublamina-b stratifying cells (Fig. 2b, c). Furthermore, the soma diameter (Fig. 2d), dendritic diameter (Fig. 2e), and pattern of Sholl crossings (Fig. 2b) measured in the sublamina-a stratifying cells are identical to M1 ipRGCs localized in the dorsal retina of the *OPN4^Cre;Brn3b^zDTA* (Fig. b–e; cyan) and consistent with previous studies of M1 type morphology[33,38,39]. Therefore, these *GlyT2^Cre* ipRGCs represent a mixed population of M1 ipRGCs and non-M1 ipRGCs.

To determine the spatial location of *GlyT2^Cre* ipRGCs, we used confocal microscopy to generate distribution maps in whole-mount preparations of melanopsin antibody-stained *GlyT2^Cre;Ai140* retinae. *GlyT2^Cre*-expressing cells (Fig. 3a, b), and melanopsin-expressing ipRGCs were found across the entire retina (Fig. 3d). However, EGFP-positive ipRGCs (Fig. 3a, b) were localized to the dorsal periphery of the retina (Fig. 3e), interspersed among other dorsal ipRGCs (Fig. 3a, d, e). Their location in the dorsal retina resembles the asymmetric distribution of cone photoreceptors, more specifically, the region of the retina that

contains predominantly green cones and few UV cones[40,41]. Co-staining with the mouse s-opsin antibody that selectively labels UV opsin demonstrated *GlyT2^Cre* expressing ipRGCs were located above the UV cone transition zone (Fig. 3c, e, f), occupying the dorsal region of the retina with low UV cone density (Fig. 3g, h). This area is positioned to receive light from below the horizon[42], suggesting these ipRGCs may encode luminance reflected from the ground (Fig. 3g, h).

Subsequent experiments that stained *GlyT2^Cre;Ai9* retina with the RGCs-specific antibody RBPMS (Fig. 3i) and whole retina mapping (Fig S2) demonstrated that GlyT2 ipRGCs accounted for ~12% of all ipRGCs (Fig. 3j; 275 ± SEM 9.4 ipRGCs of 3018 ± SEM 244 ipRGCs; $n = 9$ and 3 retina from 6 and 3 mice). We also demonstrated that 99.4% of RBPMS and TdTomato-positive cells were melanopsin-positive (Fig. 3k and S2) suggesting all *GlyT2^Cre* RGCs are ipRGCs. While most of the non-M1 ipRGCs appear to be M2 ipRGCs, staining the retina with the SMI-32, a selective marker of alpha RGCs revealed that some of the non-M1 ipRGCs likely belong to the M4 subtype (Fig. S3; <4% of GlyT2 ipRGCs; $n = 14 ± 2.78$ cells from $n = 3$ retina in $n = 3$ animals)[36,37].

Previous studies report that M1 ipRGCs are denser in the dorsal retina[3,43–45], so we reasoned that *GlyT2^Cre*-expressing M1 ipRGCs might account for this asymmetry. If our hypothesis is correct, the density of

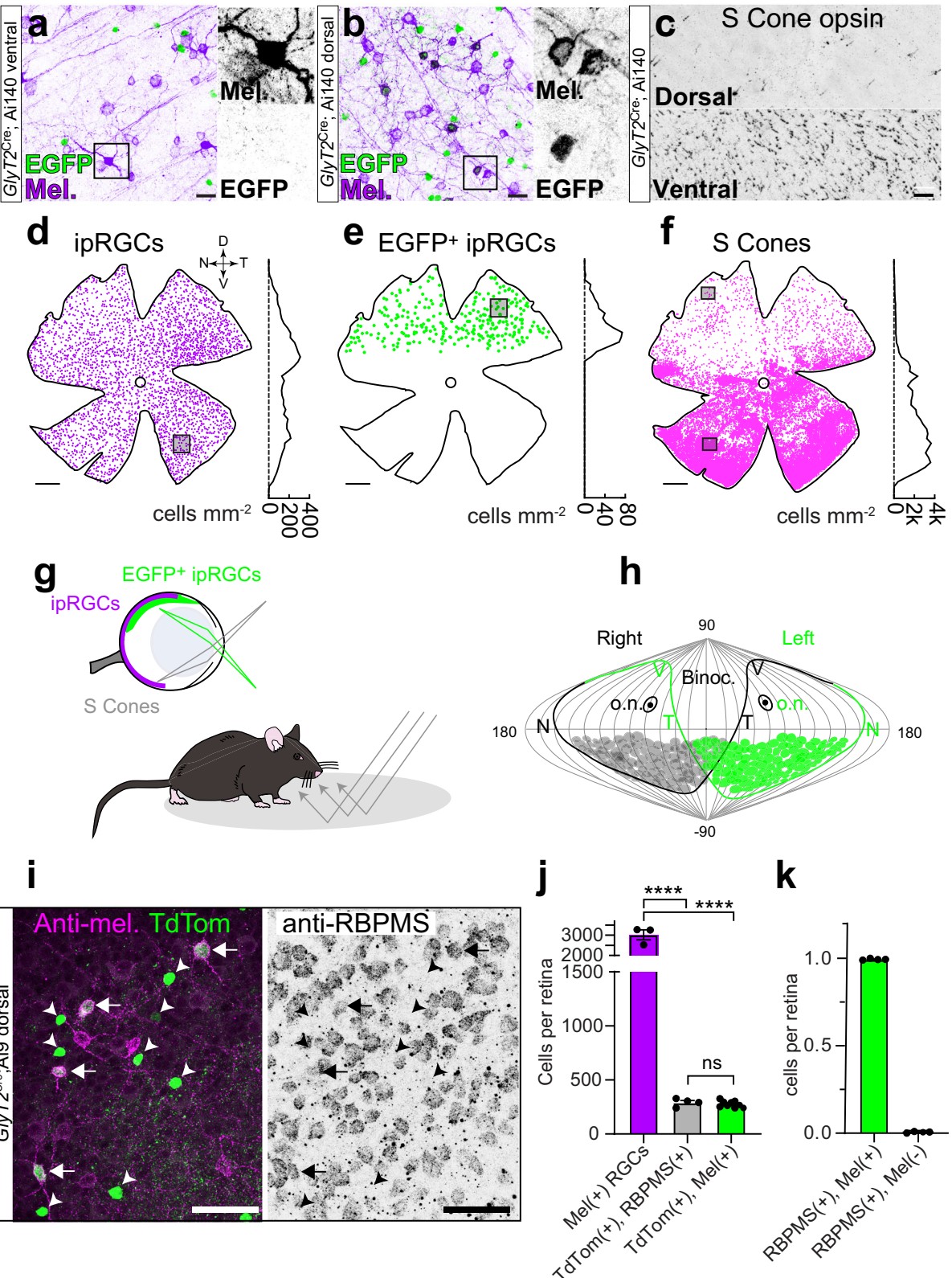

M1 ipRGCs in the dorsal and ventral retinas should be the same if we discount the *GlyT2^{Cre}* M1 ipRGCs. To test this hypothesis, we examined the retinal distribution of all M1 ipRGCs using confocal microscopy of melanopsin stained *GlyT2^{Cre};Ai140* retina taking advantage of their stratification in the IPL (Fig. 4a–c) and high melanopsin expression (Fig. 4c)[28,33]. Whole retina density maps of M1 ipRGCs (*n* = 698 ± 30.8 cells; *n* = 5 retinae from *n* = 3 animals) confirm their increased dorsal

density (Fig. 4d, i)[44] and that they were evenly interspersed with *GlyT2^{Cre}* M1 ipRGCs (Fig. 4c, *n* = 125 ± 3.9 cells; *n* = 5 retinae from *n* = 3 animals) in approximately equal percentage to *GlyT2^{Cre}* non-M1 ipRGCs (Fig. 4g). When we subtracted *GlyT2^{Cre}* M1 ipRGCs (Fig. 4f, h–k), the quantity and density of M1 ipRGCs between the dorsal and ventral retina were equivalent, confirming our hypothesis (Fig. 4i–k and S5). Dorsal *GlyT2^{Cre}* ipRGCs did not overlap with RGCs labeled in mice

**Fig. 3 | Localized distribution of ipRGCs capture ground luminance.**
**a**–**f** Confocal micrographs and density maps of neurons labeled in *GlyT2^Cre* mice. **a** Melanopsin-positive ipRGCs in the ventral retina are negative for EGFP. **b** In the dorsal retina, many ipRGCs are EGFP-positive and overlap with EGFP-negative ipRGCs. **c** S-cone opsin staining illustrates sparse labeling in the dorsal retina and dense staining in the ventral retina. **d** Location map of melanopsin immuno-positive ipRGCs (*n* = 3314). Dorsal-ventral density is plotted on the right (100 μm bins across the *Y* axis). Square = region shown in **a**. D dorsal, V ventral, N nasal, T temporal. **e** 293 ipRGCs were EGFP-positive and restricted to a region of the retina that is low in (**f**) S-cone opsin density. Squares = regions in **b** and **c** above. **g** Illustration of the *GlyT2^Cre* ipRGCs in the dorsal retina of the mouse encoding light in the ventral visual field compared to all ipRGCs. S-opsin-containing cones encode the upper visual field. Mouse printed with permission from © 2023 The Jackson Laboratory. **h** Distribution of *GlyT2^Cre* ipRGCs in both eyes plotted in a sinusoidal projection adapted from Bleckert et al. 2014. o.n. optic nerve. **i** Confocal micrograph of melanopsin, TdTomato, and RBPMS co-staining in the dorsal *GlyT2^Cre;Ai9* mouse retina. The left arrows indicate RBPMS-positive GlyT2Cre ipRGCs. All melanopsin-negative, TdTomato-positive cells are RBPMS negative (arrowheads). **j** Quantification of total melanopsin-positive cell bodies (*n* = 3 retina from *n* = 3 animals), cell bodies co-labeled with TdTomato & RBPMS (*n* = 4 retina from *n* = 4 animals), and cell bodies co-labeled with TdTomato and melanopsin (*n* = 9 retina from *n* = 6 mice; ****$p$ < 0.0001; ns $p$ = 0.9981). **k** All *GlyT2^Cre*-positive ipRGCs are melanopsin-positive. Percentage of TdTomato-positive and RBPMS-positive cell bodies that are melanopsin-positive (left) compared to melanopsin-negative (right) identified from whole-mount *GlyT2^Cre;Ai9* retinas (*n* = 4 retina). Data were presented as mean values ± SEM. Statistical significance was assessed using one-way ANOVA using Šídák's multiple comparisons test. Scale bars in **a** = 20 μm, **b** = 20 μm, **c** = 25 μm, **d**–**f** = 0.5 mm, **i** = 50 μm. Source data are provided as a source data file.

where *Cre* recombinase is driven by the peptide pituitary adenylate cyclase-activating polypeptide (PACAP; Fig. S4), which are abundant in the dorsal rat retina[43]. These results suggest the mouse visual system dedicates an additional M1 ipRGC visual channel to the ventral visual field. We reasoned this anatomical segregation in the retina might be mirrored in their central axonal projections, which are segregated in previously identified subtypes of M1 ipRGCs, underlying distinct behavioral functions[10,15,17].

### *GlyT2^Cre* ipRGCs innervate the outer core of the SCN

To determine the central axon projection sites of *GlyT2^Cre* ipRGCs we performed intravitreal eye injections of *Cre*-dependent AAV (Fig. 5 and Figs. S6–S11). Retina were quantified to ensure that transfected *GlyT2^Cre* ipRGCs exhibited a similar distribution and quantity of M1 and non-M1 ipRGCs found in the *GlyT2^Cre;Ai140* line (Fig. S6). To provide anatomical reference, we labeled all RGC axons with CTB eye injections (Fig. S9a, d, g, j) and eye injections performed in *OPN4^Cre* mice, where *Cre* is expressed in all ipRGCs (Fig. S9c, f, i, l).

Like previously described ipRGCs, *GlyT2^Cre* ipRGCs project to the SCN; however, this projection is unique for several reasons. First, their axonal projections to the SCN avoid a central core region and are concentrated at the ventral and lateral regions (Fig. 5a–h). Staining the SCN with an antibody against arginine vasopressin (AVP) in *GlyT2^Cre;Ai32* mice, illustrates that their axons did not project to the classically defined shell of the SCN, which is delineated by AVP neurons[46–51] (Fig. 5c–f and Fig. S7). Rather, they project to a subregion of the core, which we have named the outer core (Fig. 5c, d). In the anterior SCN, their axon terminals were located ventrally (Fig. 5a, b, d and Fig. S7a) in a region associated with neurons expressing vasoactive intestinal peptide[46]. At the most caudal region of the SCN, their axon terminals formed a lateral band with excursions outside of the SCN into the anterior hypothalamus and lateroanterior hypothalamus (Fig. 5d, e, i and Fig. S7d). This was quantified using normalized fluorescence values (F/Fo) in the brains from *n* = 4 *GlyT2^Cre* mice injected unilaterally with *AAV2-FLEx-EGFP* (Fig. 5f, g), illustrating they projected bilaterally to the SCN (F/Fo = 2.21 ± 0.27 ipsi. vs. 2.35 ± 0.33 contra $p$ > 0.999). We also quantified the cross-sectional fluorescence of axons in the SCN of mice co-injected with CTB (Fig. 5h). These localized projections suggest that *GlyT2^Cre* ipRGCs likely contribute a distinct functional role in the light-entrainment of circadian rhythms.

Outside of the SCN, the *GlyT2^Cre* ipRGCs innervated the SON, which contains neurons expressing AVP and oxytocin (Fig. 5i–l) and is thought to be involved in systemic fluid homeostasis[52], parturition[53], appetite[54,55], and social behaviors[56,57]. It is also thought that the innervation of this region is exclusively from Brn3b + M1 ipRGCs[4,6,15]. *GlyT2^Cre*-ipRGC axons most prominently innervated the region of the SON immediately dorsal to the optic tract and dorsomedial to the SON known as the perinuclear zone (Fig. 5i–k; pSON)[4,15,58]. Some of their axons did, however, innervate the SON in addition to extending medially into the lateral hypothalamus (Fig. 5j and Fig. S8a). Quantification using normalized fluorescence values reveal that innervation is primarily to the contralateral hemisphere of the pSON (Fig. 5l), unlike the bilateral innervation of the SCN (Fig. 5g) (F/Fo = 1.25 ± 0.0.04 ipsi. vs. 1.71 ± 0.07 contra.; $p$ = 0.0012, *n* = 4 animals)

Outside of the hypothalamus, *GlyT2^Cre* ipRGCs also project to the zona incerta (Fig. S8b)[4] and densely innervated the contralateral intergeniculate leaflet (IGL) (F/Fo > 1.5; F/Fo = 1.17 ± 0.17 ipsi. vs 2.35 ± 0.36 contra.; $P$ = 0.009, *n* = 6 animals), a known site of accessory circadian function[59] (Fig. 5m–o). Their axons are also projected to the contralateral vLGN, a region that influences a number of sensory and behavioral states, including fear[60]. Interestingly, *GlyT2^Cre* ipRGCs avoided the magnocellular (mc) region of the vLGN, which receives dense innervation from the retina (Fig. 5m, n), instead innervating the parvocellular division (pc), which is thought to receive very little direct retinal input[61,62] (Fig. 5m–o). Given the minimal CTB labeled axons in the pc vLGN, it is possible the *GlyT2^Cre* ipRGCs represent a dominant input to this region.

Brain regions such as the olivary pretectal nucleus (OPN) and superior colliculus are additional sites of well-established ipRGC innervation and function[17]. However, *GlyT2^Cre* ipRGCs exhibited minimal projections to these regions. In the OPN, axons showed only sparse innervation to the ventral shell of the OPN (Fig. S10a, b; F/Fo < 1.5; F/Fo = 1.42 ± 0.17, *n* = 5 animals), suggesting they have minimal involvement in the pupillary light reflex (Fig. S9a–c). Similarly, only sparse fibers were observed throughout the superior colliculus and stratum opticum (SO; Fig. S9d; F/Fo -1.0). Together these results suggest that *GlyT2^Cre* ipRGCs may exhibit their primary influence on circadian or homeostatic functions through their predominant innervation of the SCN, SON, and IGL.

### SON-ipRGCs: a mosaic of ipRGCs retro-labeled from the SON

Due to the heavy innervation of the pSON and surrounding areas when compared with previous reports, we hypothesized that (1) *GlyT2^Cre* ipRGCs may be the sole projection to this region and (2) that only the M1 morphological type of *GlyT2^Cre* ipRGCs project to this region[4,63]. Since it is comparatively isolated from the SCN and the LGN, we decided to selectively target M1 *GlyT2^Cre* ipRGCs using retrograde injections of *Cre*-dependent AAV injected into the pSON (Fig. 6a, b). These injections labeled melanopsin-positive M1 ipRGCs in the retina with dendrites in sublamina-a, restricted to the dorsal hemisphere and in similar density to those identified in *GlyT2^Cre;Ai140* (Fig. 6c, e, g and Fig. S12a; 113 ± 5.4; *n* = 2 mice). Next, we performed the same injections in *OPN4^Cre* mice, which expresses *Cre* in all ipRGCs, to determine if the dorsal location and stratification of these SON-labeled ipRGCs is specific to neurons expressing *GlyT2^Cre* (Fig. 6d–g). Significantly, the majority of ipRGCs labeled with these injections were sublamina-a-stratifying M1 ipRGCs (-97%; Fig. 6e, f and Fig. S12b) and located in the dorsal retina in similar quantity and distribution to those labeled in *GlyT2^Cre* mice (*n* = 131 ± 16.4 M1 ipRGCs, *n* = 4 ± 1

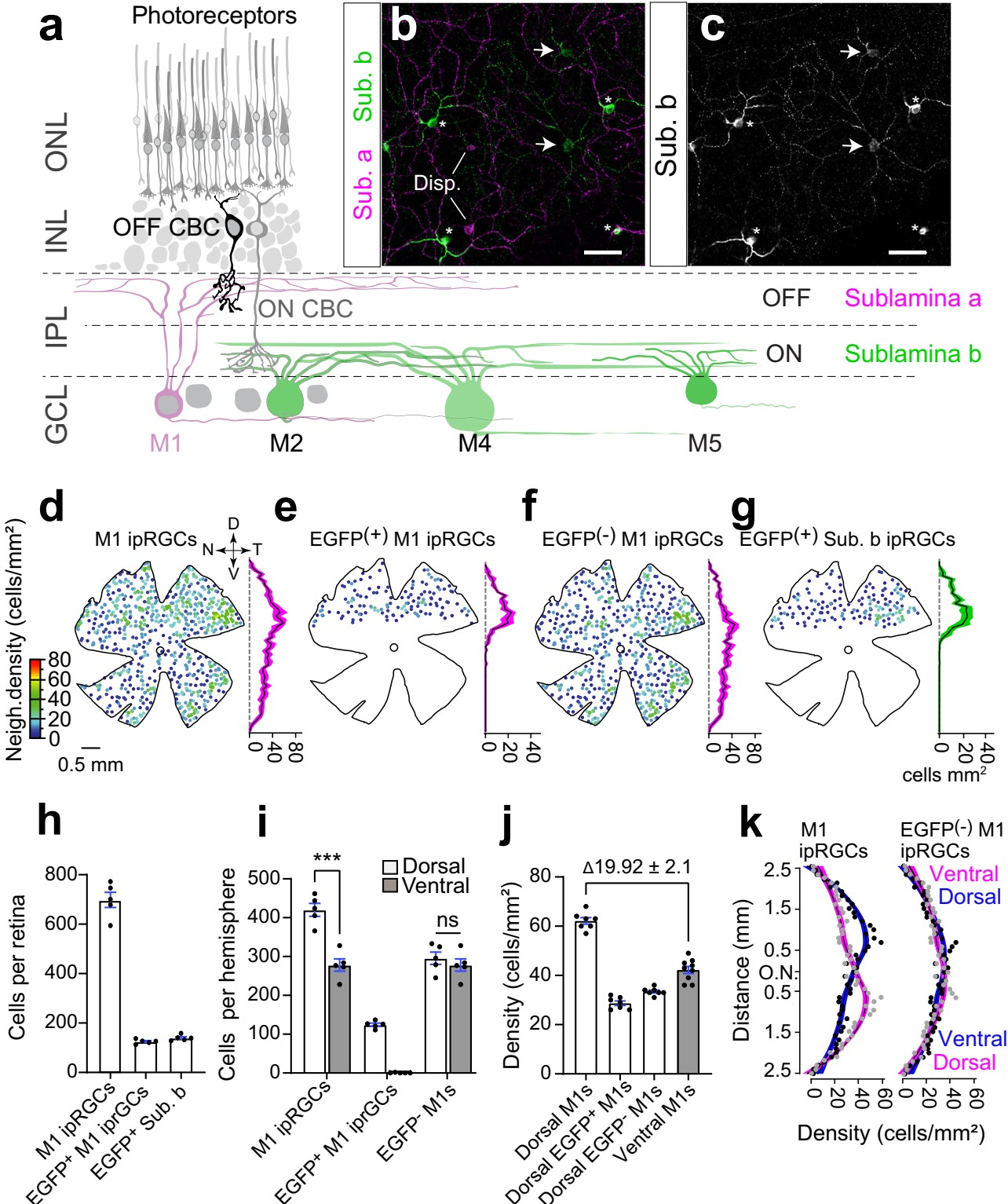

non-M1 ipRGCs, $n = 3$ mice) and similar in number to those quantified from our counts of EGFP and melanopsin-positive M1 ipRGCs in *GlyT2^{Cre};Ai140* mice (Fig. 4h).

We observed a small number of ventral M1 ipRGCs labeled by AAV injections into the SON in *OPN4^{Cre}* mice, in addition to a small number of Non-M1 ipRGCs (Fig. 6e–h and Fig. S12; $n = 27 \pm 5.5$ ipRGCs, $n = 3$ animals). As these ipRGCs were (1) rarely labeled, (2) restricted to small regions, and (3) the non-M1 ipRGCs were also predominantly in the ventral retina, we conclude that this is most likely due to spillover of

AAV into the optic tract which lies immediately ventral to the SON. We also noticed some non-ipRGCs, which appeared to be amacrine cells labeled in the retina (*GlyT2^{Cre}*, $2 \pm SD\ 2.6$ neurons from 3 mice; *OPN4^{Cre}*, $5.3 \pm SD\ 5$ neurons from 3 mice). We conclude that their labeling likely arose from trans-synaptic labeling, or viral spillover from ipRGCs in the retina, as they do not have axons passing out of the retina. Together these data suggest that the dorsal *GlyT2^{Cre}* ipRGCs may represent the sole retino-recipient projection to the SON, and further strengthen our conclusions from anatomical mapping data illustrating these ipRGCs

**Fig. 4 | GlyT2^Cre ipRGCs account for the increased dorsal ipRGC density.**
**a** Schematic of the retinal circuit with rod and cone photoreceptors located in the outer nuclear layer (ONL) and ipRGCs in the ganglion cell layer (GCL). Some of the six ipRGC types differ in their dendritic size and stratification in sublamina-a and sublamina-b. **b**, **c** Confocal micrographs of melanopsin antibody-stained ipRGCs illustrating M1 ipRGCs (asterisks) are identified by their dendrites transitioning from sublamina-b and branching in the sublamina-a of the IPL. Non-M1 ipRGCs branch earlier in sublamina-b (arrows). Disp displaced ipRGCs. **d** Neighbor density maps of morphologically identified M1 ipRGCs, **e** EGFP-positive M1 ipRGCs, **f** EGFP-negative M1 ipRGCs, and **g** EGFP-positive non-M1 ipRGCs. Each cell is color-coded according to the number of neighboring cells within a radius of 220 μm. Hotter-colored cells lie within regions of higher density, such as the dorsal retina for all M1 ipRGCs and the temporal retina for EGFP-negative M1 ipRGCs. The average density (±SEM magenta shading) at each retinal location along the dorso-ventral axis is plotted right (100 μm bins across the Y axis, $n = 5$ retina from $n = 3$ animals). **h** Bar graph of quantified M1 ipRGCs, EGFP-positive M1 ipRGCs, and sublamina-b stratifying EGFP-positive ipRGCs from whole-mount retinas ($n = 5$ retina from $n = 3$ animals). **i** Cells per hemisphere ($n = 5$ retina from $n = 3$ animals; ***$p = 0.0002$; ns $p = 0.0535$) and **j** density (dorsal vs. ventral) for morphologically identified M1 ipRGCs per 1 mm$^{-2}$ across $n = 7$ dorsal regions and $n = 9$ ventral regions within $n = 5$ retina from $n = 3$ animals. **k** Distribution of average densities along the dorsal-ventral axis for M1 ipRGCs (left) and M1 ipRGCs minus EGFP-positive M1 ipRGCs (right; $N = 5$ retina from $n = 3$ animals). Magenta and blue curves are identical per plot but inverted across the optic nerve (ON) to visually compare the distribution of average densities in the dorsal and ventral regions $n = 5$ retina from $n = 3$ animals. Data were presented as mean values ± SEM. Statistical significance was assessed using one-way ANOVA using Šídák's multiple comparisons test. Scale bar in (**b**) and (**c**) = 50 μm. Source data are provided as a source data file.

represent a distinct subtype that is located solely in the dorsal retina. Because these ipRGCs are selectively labeled from the SON, we now refer to them as SON-ipRGCs.

We noticed that SON-ipRGCs were evenly spaced in our anatomical mapping experiments, and the territorial mosaic distribution of RGCs is one of the defining characteristics of a unique functional subtype. Our retro-labeling of SON-ipRGCs with brain injections into GlyT2^Cre and OPN4^Cre mice was even more striking. The dendrites of SON-ipRGCs in the dorsal retina formed non-overlapping territorial mosaics, reminiscent of other territorial RGC subtypes (Fig. 7a–c)[23,24]. Upon close examination using confocal microscopy, the dendrites of SON-ipRGCs overlapped with the dendrites of other M1 ipRGCs in sublamina-a of the IPL stained with anti-melanopsin, and displaced M1 ipRGCs somata (Fig. 7d). SON-ipRGCs have a uniform and unique dendritic morphology with 3–4 short dendritic segments that project through sublamina-b and extend their terminal dendrites into sublamina-a (Fig. 6c). To provide a quantitative framework of analysis of the mosaic distribution of SON-ipRGCs, we quantified (1) the density recovery profile[64,65], a measurement of cell density at increasing distances from the soma (Fig. 7e, g and Fig. S13a, b, d), (2) regularity index, a measurement of even soma distribution using the deviation of nearest neighbor measurements (Fig. 7g, h) and (3) the coverage factor, which is a quantitative measurement of dendritic overlap in mosaic distributions (Fig. 7i and Fig. S13c). Because ipRGC distributions are inherently sparse, we also performed analysis on modeled random distributions of cells matched to the retro-labeled SON-ipRGCs for soma size and density (Fig. 7f–h).

Our density recovery profile data indicated that unspecified M1 ipRGCs and non-SON-projecting M1 ipRGCs, as well as our random modeled distribution (matched for soma size and density) each overlapped significantly, as evidenced by high values in very close proximity to the soma (<100 μm; Fig. 7e, f). SON-ipRGCs labeled with retro-injections in GlyT2^Cre and OPN4^Cre mice exhibited a clearly defined exclusion zone around the soma, which indicates their cell bodies may be regularly spaced (Fig. 7e, f and Fig. S13a, b), although the presence of an exclusion zone is not conclusive evidence that a retinal mosaic is regular. We next calculated nearest neighbor (NN) regularity indexes for each SON-labeled retina and the matched random modeled distribution by computing the ratio of the mean nearest neighbor (NN) distance (Fig. 7g) to its standard deviation per retina. The random modeled distribution had a ratio of 1.79 ± 0.14 (Fig. 7h), consistent with a theoretical random distribution (NN regularity indexes <1.91[66]). SON-labeled retina had a ratio of 3.05 ± 0.27 (Fig. 7h), similar to other retinal cell types with mosaic distributions (NN regularity indexes >1.91)[67].

We next examined their coverage factor, which measures the average number of dendritic fields within an RGC mosaic overlapping any point in space. Most RGC subtypes that represent a functional visual channel have a coverage factor of ~2, indicating there are roughly two dendritic fields (or receptive fields) at any point in the

retina[68]. To calculate the coverage, we used the average dendritic field diameter from morphological Neurobiotin fills (324 ± 18 μm), as the edges of the dendritic fields labeled from SON virus injections were difficult to resolve due to their overlap. Using these measurements, we found that SON-ipRGCs had a coverage factor of just over 2 (Fig. 7i; 2.2 ± 0.18 GlyT2^Cre; 2.1 ± 0.03 OPN4^Cre), indicating each point in the dorsal retina is covered by at least 2 SON-projecting ipRGCs. The coverage factor of Non-SON-projecting ipRGCs was 3.6 ± 0.19, and all M1 ipRGCs was 5.7 ± 0.16 (Fig. 7i, Fig. S13c, and Supplementary Table 2). This indicates that SON-ipRGCs are territorial, and provide seamless coverage of the retina with minimal overlap, similar to some other highly territorial RGC subtypes[24]. These results also support the hypothesis that there may be either two more territorial M1 ipRGCs subtypes in the dorsal retina or an additional single subtype of M1 ipRGCs with higher coverage and slightly more overlap than SON-ipRGCs. Together, these results strongly support the hypothesis that SON-ipRGCs are one of multiple M1 ipRGC subtypes in the dorsal retina.

Next, we examined other central projection locations following SON injections (Figs. S14, S15, S16). As these targeted injections (Fig. S14a, b) label predominantly M1 ipRGCs, and account for most GlyT2^Cre M1 ipRGCs, this allowed us to determine the collateral projection patterns of SON-ipRGCs alone, without non-M1 ipRGC projections (Fig. 5 and Figs. S6, S9, S10, S11). We compared the innervation of SON-ipRGCs labeled in the GlyT2^Cre and OPN4^Cre with the anterograde projections observed in the GlyT2^Cre line. SON-ipRGC collaterals avoided the central core of the SCN (Fig. S14c–f) and the pattern of collateral innervation to the SCN was identical between the GlyT2^Cre and OPN4^Cre cohorts (Fig. S14e–g). These results illustrate that the unique projections to the outer core of the SCN are from SON-ipRGCs. Projections to the IGL and parvocellular layer of the vLGN were also observed (Fig. S15a, b), however, these projections were substantially reduced compared to anterograde labeling (Fig. S15c). The projection to the ventral part of the dLGN was absent (Fig. S15a, b, d). These results suggest that the vLGN and dLGN are primarily innervated by non-M1 GlyT2^Cre ipRGCs labeled in the eye injections (Fig. 5l–n and Fig. S6). Projections to the OPN were sparse in both the GlyT2^Cre and OPN4^Cre animals (Fig. S16) consistent with the minimal innervation pattern quantified in the anterograde labeling (Fig. S10). Together these results suggest the primary projections of the SON-ipRGCs are to the SON, SCN, and IGL.

## SON-ipRGCs release glutamate at central synapses

Having established that SON-ipRGCs represent a unique subtype of M1 ipRGCs according to their expression and distribution in the retina, we asked if their targeting in GlyT2^Cre mice underlies unique neurotransmitter release in the brain. This is particularly important given the recent discovery that some ipRGCs, which project to the SCN, IGL, and OPN, release GABA at their central synapses[69]. As SON-ipRGCs are labeled in a mouse line that selectively labels inhibitory neurons

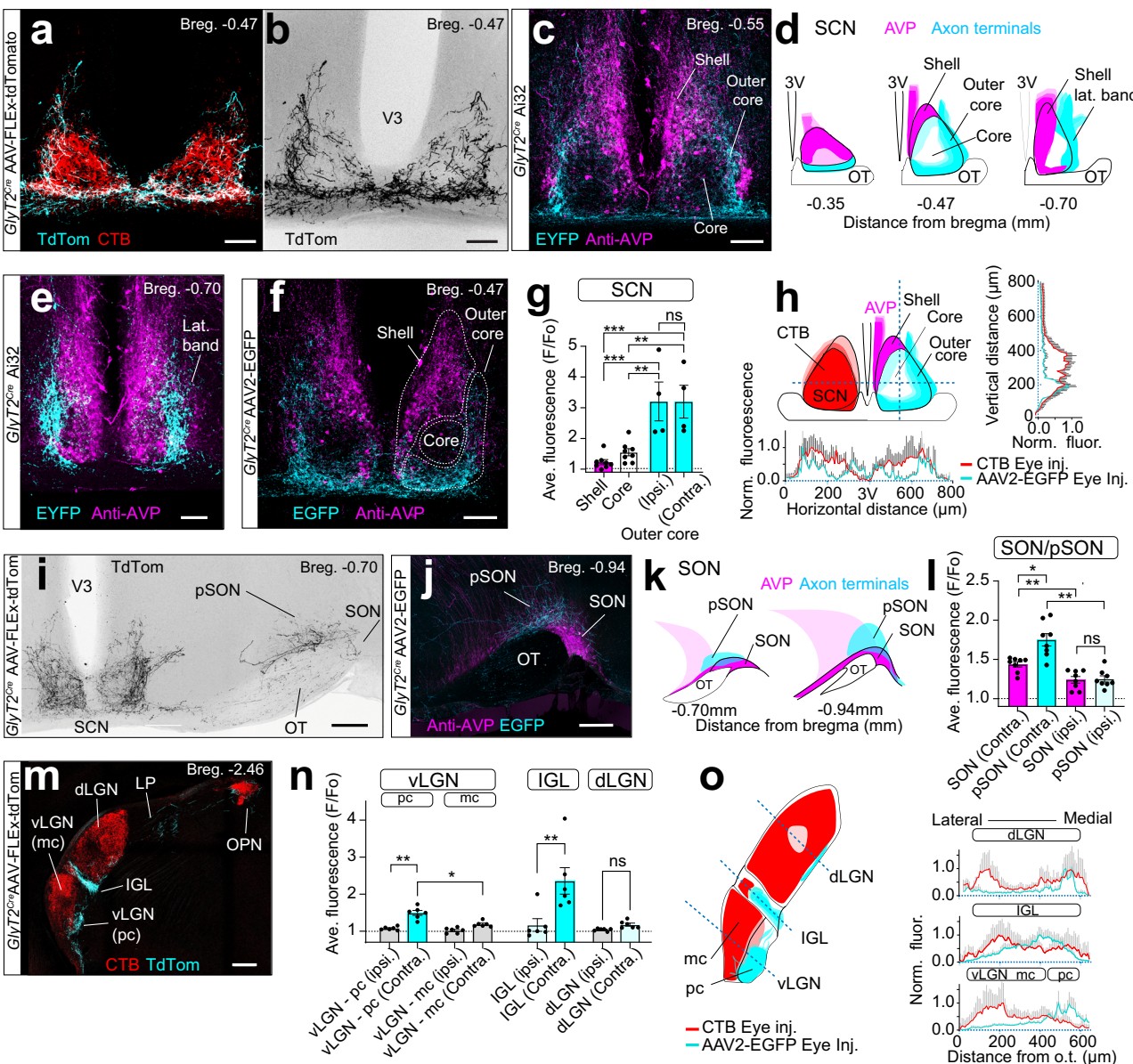

**Fig. 5 | Axon projections to an SCN subregion and to the pSON. a, b** Eye injections of *Cre*-dependent AAV in the *GlyT2^Cre* mouse illustrating GlyT2Cre-ipRGC axons avoid the central core, which is stained by cholera toxin B (CTB). **c** Axons avoid the shell of the SCN localized by arginine vasopressin neurons. **d** Rostral (bregma −0.35) to caudal (bregma −0.70) distribution of axons. **e** Axons in the caudal SCN form a lateral band. **f** SCN, shell, core, and outer core regions used for fluorescence quantification following *AAV2-FLEx-EGFP* eye injections. **g** Normalized fluorescence (F/Fo) of the SCN core and shell; bilateral (contralateral and ipsilateral), and SCN outer core for the ipsilateral and contralateral SCN (*n* = 4 animals, data were presented as mean values ± SEM; ***top *p* = 0.0004; ** top *p* = 0.003; ***bottom *p* = 0.0004; **bottom *p* = 0.0028). **h** Illustration (left) and fluorescence (right and below) through the horizontal and ventral axis of SCN (*AAV2-FLEx-EGFP*), *n* = 6 animals, vs. CTB eye injections, (*n* = 2 animals). **i** Axons extend into lateral hypothalamic (LH) and anterior hypothalamic (AHC) areas, and there is a strong

projection to the episupraoptic nucleus (pSON). **j** Axon innervation of the pSON and SON above the optic tract (OT) localized with AVP staining. **k** Rostral and caudal SON innervation. **l** Normalized fluorescence values of the SON and pSON measured at Bregma = −0.7 and −0.94 (*n* = 4 animals; error bars = ±SEM; **p* = 0.0149; **top *p* = 0.003; **bottom *p* = 0.0012; ns *p* = 0.9973). **m** Axons are innervating the parvocellular (pc) vLGN, LP, IGL, and OPN. **n** F/Fo of the vLGN, IGL, and dLGN (*n* = 4 animals; data were presented as mean values ± SEM; **p* = 0.0189; **left *p* = 0.0049; **right *p* = 0.0099; ns *p* = 0.0825). **o** F/Fo through the contralateral vLGN, IGL, and dLGN for *AAV2-FLEx-EGFP* vs. CTB eye injections in *GlyT2^Cre* mice. Profiles illustrate *GlyT2^Cre* ipRGC innervation to the pc region of the vLGN, medial IGL, and ventral dLGN (*AAV2-FLEx-EGFP* *n* = 6 animals; CTB eye injections *n* = 2 animals). Statistical significance was assessed using one-way ANOVA with Šídák's multiple comparisons test. Scale bars in **a**–**c**, **e**, **f** = 100, **l**, **j** = 200 μm, **m** = 250 μm. Source data are provided as a source data file.

throughout the brain and retina, we asked if they released GABA or glycine using two optogenetic approaches to express ChR2 in their axon terminals and to record light-evoked neurotransmitter release (Figs. 8, 9). We chose to record from multiple central locations to rule out the possibility SON-ipRGCs differentially release neurotransmitters at different central locations. Our recordings were focused primarily on the SCN, SON, and IGL, due to their dense innervation from SON-ipRGCs.

We crossed *GlyT2^Cre* mice with a *Cre*-dependent ChR2^EYFP reporter (Ai32; Jackson 024109), which resulted in the expression of ChR2 in the axons and nerve terminals of SON-ipRGCs (Fig. 8a–d). Lateral SCN neurons were targeted in coronal slices (Fig. 8b, c). "Prior to break-in, in cell-attached voltage-clamp mode, photo-stimulation activated robust extracellular unclamped action potential (AP) currents, which demonstrates that the SCN neuron was depolarized beyond its action potential threshold by the release of an excitatory transmitter.

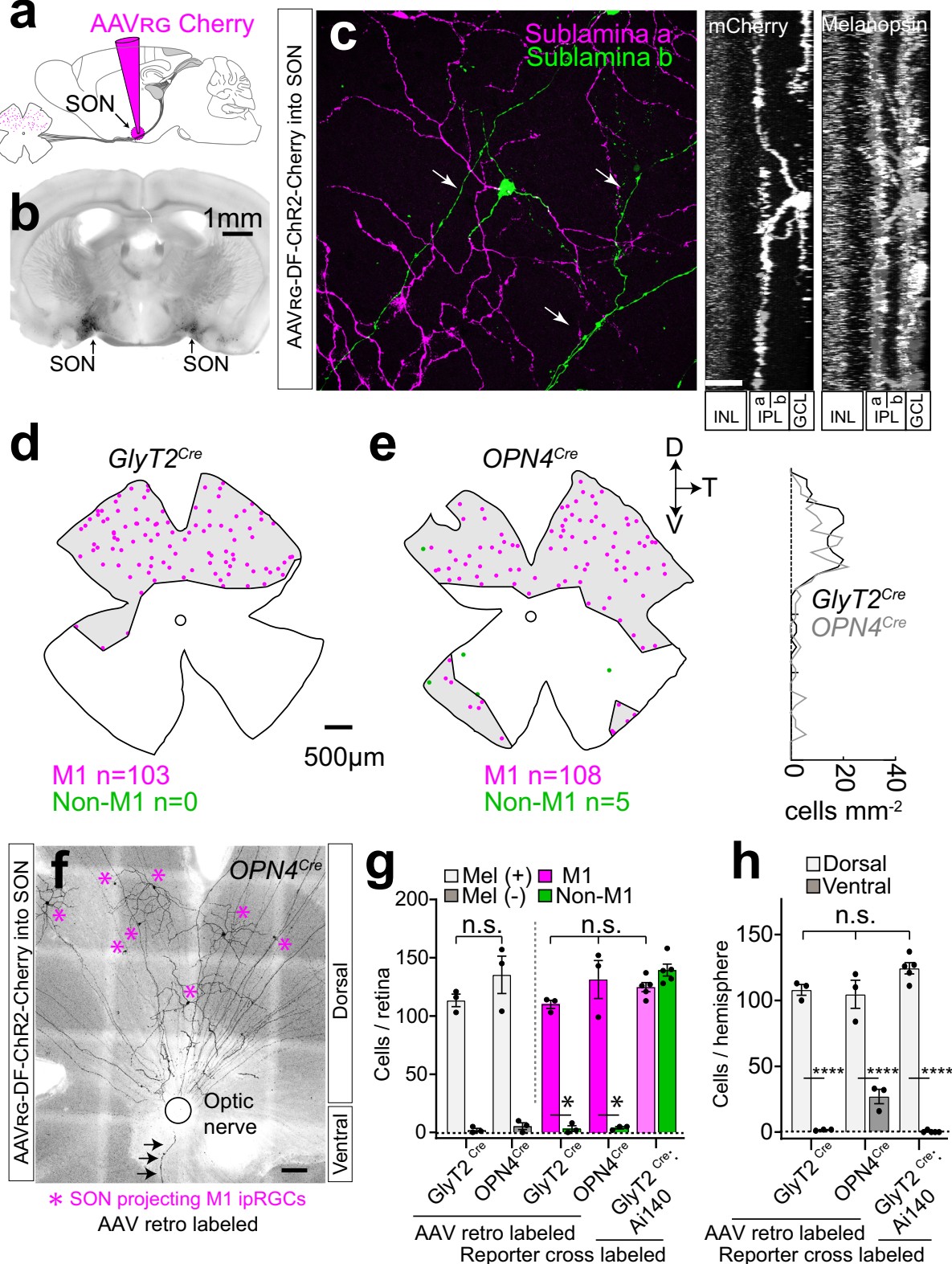

The amplitude and latency of AP currents (Fig. 8e) were robust and fast, consistent with monosynaptic excitatory synaptic connections (mean ± SEM 7.2 ± 0.5 ms (range 6.3–8.2 ms) and 193.7 ± 74.2 pA (range 53.8–306.3 pA), $n = 3$) and were blocked by CNQX and AP-5 (Fig. 8e, j; $n = 2$). In whole-cell voltage-clamp recordings, we detected photo-stimulation evoked inward post-synaptic currents (PSCs) at holding potentials between −60 and −40 mV (Fig. 8f). The PSCs latency and

amplitude were 6.7 ± 0.2 ms (range 6.3–7.4 ms) and 35.2 ± 10.0 pA (range 15.4–71.3 pA, $n = 5$, Fig. 8g, h). Similar excitatory synaptic input to SCN neurons was demonstrated by electrical stimulation of the optic chiasm (Fig. 8b, f–h), which resulted in larger and faster PSCs (Fig. 8f, h). Pharmacological blockers were used to identify the neurotransmitter released by SON-ipRGCs. The glycine receptor antagonist strychnine (1 μM) and the GABA$_A$ receptor antagonist SR-95531

**Fig. 6 | _GlyT2_$^{Cre}$ ipRGCs form the principle projection to the SON.**
**a**, **b** Stereotactic injections of _Cre_-dependent retroAAV into the SON labels a population of ipRGCs. **c** Confocal micrograph (left) and side z-projections (right) illustrate that retro-labeled cells are M1 ipRGCs; with dendrites that stratify in sublamina-a of the inner plexiform layer (IPL). Arrows indicate axons in the ganglion cell layer (GCL) INL = inner nuclear layer. **d** Distribution maps of retro-labeled ipRGCs in _GlyT2_$^{Cre}$ and **e** _OPN4_$^{Cre}$ mouse lines. The density at each retinal location along the dorso-ventral axis is plotted on the right. **f** Confocal micrograph illustrates dorsal retro-labeled SON-projecting M1 ipRGCs (asterisks) in the _OPN4_$^{Cre}$ mouse and a single ventral axon (arrows). **g** Bar graphs of ipRGCs per retina that are melanopsin-positive or negative (left; error bars ± SEM $p = 0.2633$) and M1 ipRGCs or non-M1 ipRGCs in SON-injected _GlyT2_$^{Cre}$ and _OPN4_$^{Cre}$ mice retinas labeled with SON injections of retroAAV (data were presented as mean values ± SEM; $n = 3$ retina from $n = 2$ _GlyT2_$^{Cre}$ animals, $n = 3$ retina from $n = 3$ _OPN4_$^{Cre}$ animals, $n = 5$ retina from $n = 3$ _GlyT2_$^{Cre}$;_Ai140_ animals; left/middle $p = 0.1577$, middle/right 0.339 and left/right 0.8762). ipRGCs identified in _GlyT2_$^{Cre}$;_Ai140_ retinas quantified for comparison. **h** Bar graphs of a population of ipRGCs per hemisphere in the SON-injected _GlyT2_$^{Cre}$ and _OPN4_$^{Cre}$ mouse retinas. _GlyT2_$^{Cre}$;_Ai140_ retinas quantified for comparison (data are presented as mean values ± SEM; $n = 3$ retina from $n = 2$ _GlyT2_$^{Cre}$ animals, $n = 3$ retina from $n = 3$ _OPN4_$^{Cre}$ animals, $n = 5$ retina from $n = 3$ _GlyT2_$^{Cre}$;_Ai140_ animals, ****$p = {<}0.0001$, ns left/middle $p = 0.9985$, left/right 0.1445, and middle/right 0.0541). Data were presented as mean values ± SEM. Statistical significance for **g**, right, and **h** was assessed using one-way ANOVA with Šídák's multiple comparisons test and for **g** left using an unpaired two-tailed $t$-test. Scale bars in **c** = 20 µm, **d**, **e** = 500 µm, **f** = 100 µm. Source data are provided as a source data file.

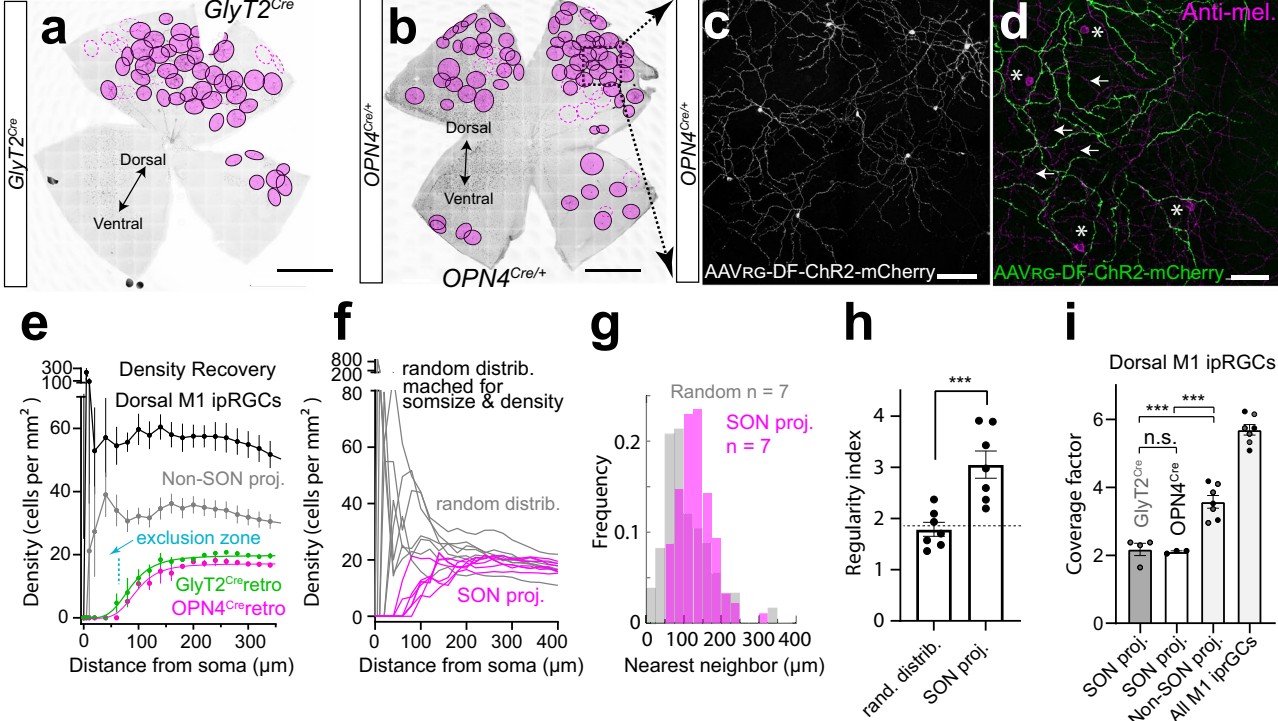

**Fig. 7 | SON-ipRGCs form a tiling mosaic that minimizes dendritic overlap.**
Whole retina images of tiling SON M1 ipRGCs retro-labeled in the _GlyT2_$^{Cre}$ (**a**) and _OPN4_$^{Cre}$ (**b**) mice with their dendritic arbor borders circled in magenta. Dashed lines represent partially labeled ipRGCs. **c** Confocal micrograph of mosaic spacing in SON M1 ipRGCs retro-labeled in the _OPN4_$^{Cre}$ mouse line. **d** Higher magnification confocal image of OFF stratifying dendrites which overlap with surrounding M1 ipRGCs dendrites stained with anti-melanopsin (arrows). Some displaced M1 ipRGC cell bodies (asterisk) are unlabeled. **e** Density recovery profile displaying the density of cell bodies at increasing distances from each soma. The exclusion zone indicates minimal overlap of SON-ipRGCs ($n = 6$ retina from $n = 5$ animals; data were presented as mean values ± SEM. **f** Comparison of individual density recovery profiles per retina with a modeled random distribution of points matched for soma size (14-µm diameter) and density ($n = 6$ retina from $n = 5$ animals). **g** Frequency distribution of nearest neighbor measurements for SON-labeled ipRGCs ($n = 7$ regions in $n = 6$ retina from $n = 5$ animals) and the matched random modeled distribution. **h** Regularity index, calculated by computing the ratio of the mean nearest neighbor distance to its standard deviation per retina, for SON-labeled ipRGCs and the matched random modeled distribution ($n = 6$ retinas from $n = 5$ animals; data were presented as mean values ± SEM; $p = 0.0063$, paired two-tailed $t$-test). **i** Coverage factor or the proportion of dendritic overlap calculated from the average diameter of M1 ipRGCs, measured from Neurobiotin fills. Non-SON proj. M1s = M1 ipRGCs not labeled in the central injection. Dorsal M1 ipRGCs = Non-SON-projecting M1 ipRGCs + retro-labeled SON-projecting M1 ipRGCs ($n = 6$ retina from $n = 2$ _GlyT2_$^{Cre}$ animals and $n = 3$ _OPN4_$^{Cre}$ animals; data were presented as mean values ± SEM; ns $p = 0.9951$; ***left $p = 0.0001$; right 0.0002). Statistical significance was assessed using Šídák's multiple comparisons test. Scale bars in **a**, **b** = 1 mm, **c**, **d** = 100 µm. Source data are provided as a source data file.

failed to inhibit photo-stimulation-induced PSCs (Fig. 8f, i). In contrast, PSCs were blocked by co-application of the selective AMPA and NMDA glutamate receptor antagonists CNQX (20 µM) and AP-5 (50 µM; Fig. 8f, i, j). Together, these results are consistent with a model where SON-ipRGCs are excitatory and release glutamate onto SCN neurons.

The whole-cell patch electrodes contained Neurobiotin and the location of recorded SCN neurons and their proximity to SON-ipRGC axon terminals was reconstructed with confocal microscopy. While only a small percentage of recordings resulted in photo-stimulation-evoked PSCs, the locations of connected neurons were mapped to each slice by referencing infrared microscopy images taken of the living slice with the subsequent post-fixed confocal images (Fig. 8k and Fig. S17). The locations of synaptically connected SCN neurons were consistent with anterograde and retrograde tracing experiments showing SON-ipRGC axon terminals resided in the outer core of the SCN (Fig. 5 and Figs. S7, S14).

Inward PSCs with a latency of 6.27 ± 0.39 ms (range 5.34–7.80, $n = 6$) and amplitude of 50.1 ± 3.4 pA (range 38.8–59.4 pA, $n = 6$)

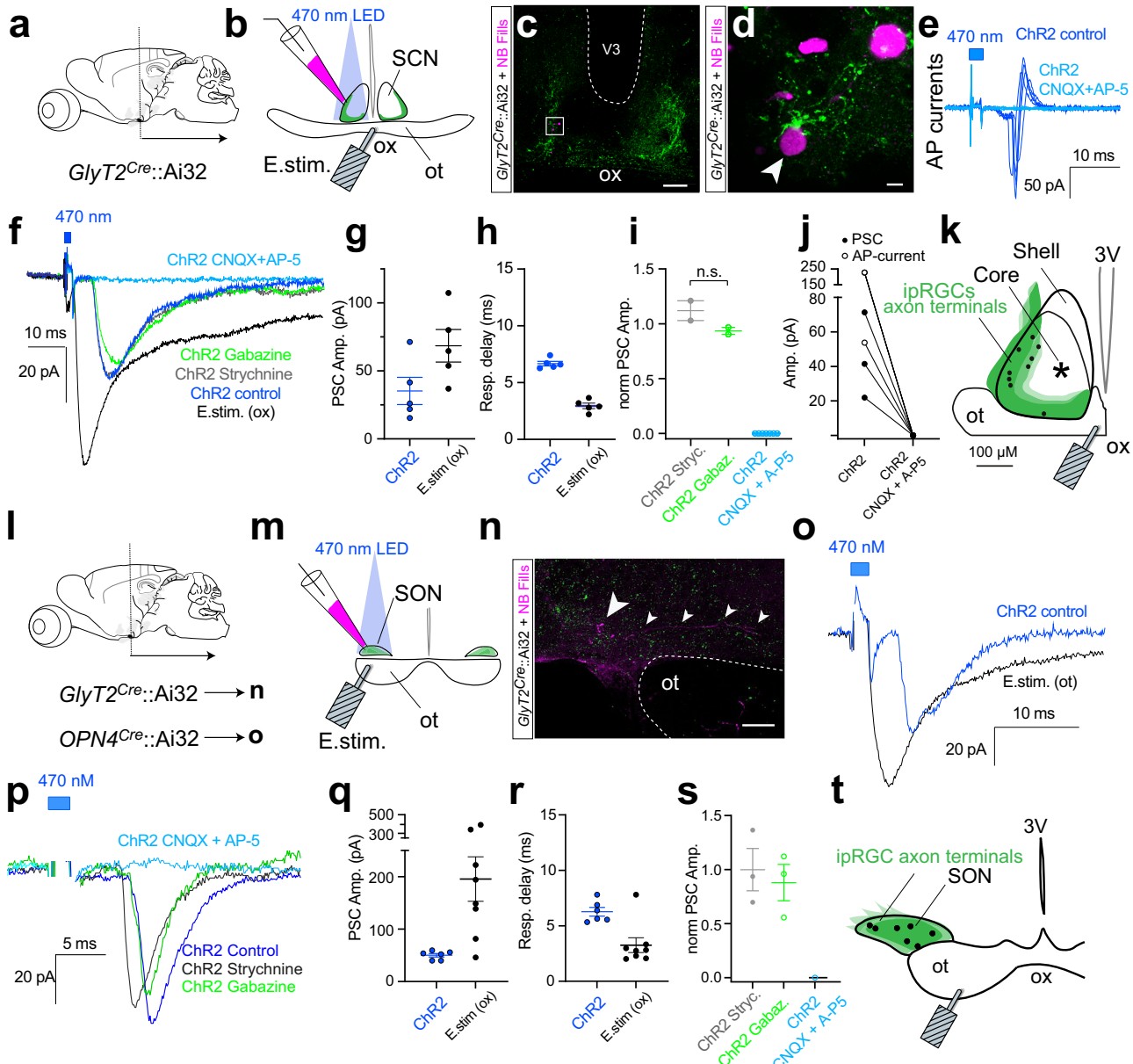

**Fig. 8 | SON-ipRGCs release glutamate in the SCN and SON. a** Brain slice recordings in the SCN performed in coronal sections of *GlyT2^Cre^;Ai32* mice. **b** Confocal images of an SCN slice, immunostained following recordings illustrating Neurobiotin-filled cells and EYFP-positive axon terminals. **d** Zoomed region from **c**. **e** Photo-stimulation-evoked action potential currents (dark blue) were abolished following in CNQX + AP-5. **f** Whole-cell voltage-clamp recordings of electrical- and ChR2-evoked inward post-synaptic currents (PSCs) in control, 1 μM strychnine, 10 μM SR-95531, and CNQX + AP-5. ChR2-evoked currents were abolished with CNQX (20 μM) and AP-5 (50 μM). **g** Response amplitude and **h** delay of ChR2- evoked (Left) and electrical-stimulation-evoked (E.Stim) PSCs in the same neurons (*n* = 5 animals). **i** ChR2-evoked response amplitude in strychnine (*n* = 2 animals), SR-95531 (*n* = 2 animals), or CNQX + AP-5 (*n* = 7 animals) normalized to the control response amplitude (data were presented as mean values ± SEM; ns *p* = 0.6917; unpaired two-tailed *t*-test). **j** PSC amplitude before and after CNQX + AP-5. **k** Photoresponsive cell locations. **l, m** Coronal recording configuration for the SON of the *GlyT2^Cre^;Ai32* and *OPN4^Cre^;Ai32* mice. **n** Confocal images of the immunostained SON following recordings. **o** Whole-cell voltage-clamp recordings of electrical- and ChR2-evoked inward PSCs in the SON. **p** Whole-cell voltage-clamp recording traces of electrical- and ChR2-evoked inward PSCs in control, 1 μM strychnine, 10 μM SR-95531, and CNQX + AP5. ChR2-evoked currents were abolished with CNQX (20 μM) and AP-5 (50 μM). **q** Response amplitude and **r** delay of ChR2- evoked and electrical-stimulation-evoked PSCs in pSON neurons (*n* = 6 animals). **s** ChR2-evoked response amplitude in strychnine (*n* = 3 animals), SR-95531 (*n* = 3 animals), or CNQX + AP-5 (*n* = 1 animal) normalized to the control response amplitude. **t** Anatomical location of photoresponsive cells. Data were presented as mean values ± SEM. Statistical significance was assessed using one-way ANOVA with Sidak correction for comparisons between multiple groups (**p* ≤ 0.01). Scale bars in **c** = 100 μm, **d** = 5 μm, **n** = 100 μm. Source data are provided as a source data file.

remained present following application of the glycine receptor antagonist strychnine (1 μM) and the GABA$_A$ receptor antagonist SR-95531 were recorded in voltage-clamped SON neurons (Fig. 8i–s). Electric stimulation of the optic chiasm evoked PSCs in SON neurons confirming the retinal projection to this nucleus (latency 3.3 ± 0.7 ms, amplitude 195.4 ± 42.4 pA, *n* = 8). Similarly, the location of each

synaptically connected neuron was mapped to the SON/pSON area in a distribution consistent with a pattern of SON-ipRGC innervation (Fig. 5).

To determine if SON-ipRGCs might differentially release neurotransmitters at separate central locations, we performed whole-cell voltage-clamp recordings in the IGL in coronal slices.

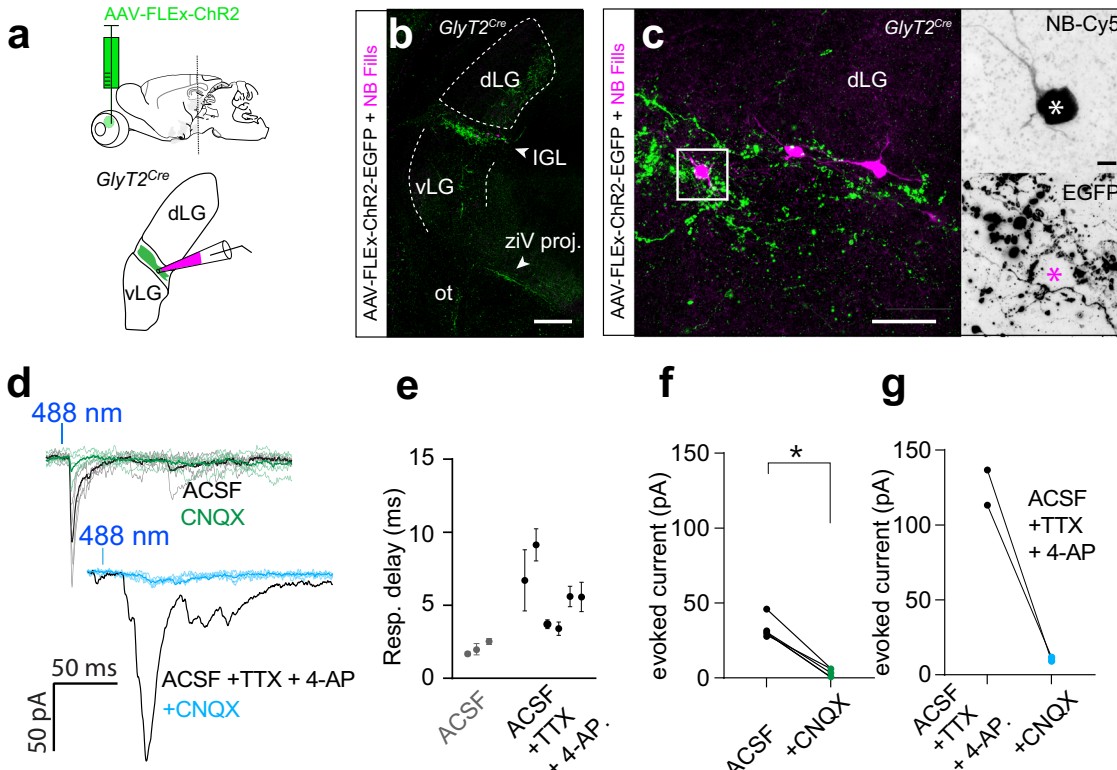

**Fig. 9 | SON-ipRGCs release glutamate in the IGL. a** Illustration of brain slice recordings in the IGL of the *GlyT2^Cre^* mouse 2–3 weeks following Cre-dependent ChR2 expression in the eye. **b**, **c** Confocal images of EGFP-expression in the IGL brain slice, fixed after recording. Biocytin-filled IGL neurons surrounded by SON-ipRGC terminals. **d** Photo-stimulation-evoked inward PSCs blocked by the excitatory antagonist CNQX. Photo-stimulation evoked inward PSCs in tetrodotoxin and 4-aminopyridine (4-AP) were blocked with CNQX. **e** Response delay indicating the time to peak following a 1 ms ChR2 stimulation in ACSF (*n* = 3 cells data were presented as mean values ± SEM), and following the application of TTX and 4-AP (*n* = 5 cells, data were presented as mean values ± SEM). **f** Amplitude of photo-stimulation-evoked PSCs in ACSF before and after CNQX (Statistical significance was assessed using a two-tailed unpaired *t*-test; \**p* = 0.0142). **g** Amplitude of photo-stimulation-evoked PSCs in ACSF with TTX and 4-AP before and after CNQX. Scale bars in **b** =;250 μm, **c** = 50 μm, **c** inset = 10 μm. Source data are provided as a source data file.

However, because the IGL receives inhibitory input from other central brain regions[62], some of which may contain neurons labeled in the *GlyT2^Cre^* mouse line (Fig. S17), we restricted ChR2 expression to the retina with eye injections of *Cre*-dependent ChR2 in *GlyT2^Cre^* mice (Fig. 9a). After allowing 2–3 weeks for the ChR2 expression, we recorded from the IGL, which was targeted in coronal slices with brief epi-fluorescent illumination to identify EGFP-expressing axon terminals, which form a dense band in the IGL (Fig. 9b). IGL neurons were filled with biocytin and recovered for confocal microscopy (Fig. 9b, c). Photo-stimulation evoked inward PSCs were completely abolished with the bath application of the AMPAR antagonists CNQX (Fig. 9d–f). These excitatory PSCs were, on average smaller than evoked in recordings from *GlyT2^Cre^;Ai32^ChR2:EYFP^* mice, which may be due to viral expression of ChR2 being lower than expression driven by the *Ai32* line. To augment photo-stimulation-induced PSCs and rule out the possibility that inhibitory synaptic terminals need greater depolarization to reach the threshold required to activate transmitter release, we included K-channel blocker 4-aminopyridine (4-AP) in the ACSF and also included tetrodotoxin (TTX) to reduce poly-synaptic events. Despite 4-AP substantially increasing the light-evoked currents, they were completely abolished by the co-application of CNQX (Fig. 9d). When we expressed ChR2 in the axon terminals of ipRGCs, we found no evidence for inhibitory synaptic release. Together our results provide direct evidence for ipRGC synaptic inputs to the SON and demonstrate that despite their targeting with the *GlyT2^Cre^* line, SON-ipRGCs release glutamate at multiple central synapses in the brain.

## Discussion

In most retinal ganglion cell subtypes, the fundamental organizing feature of retinal output is a mosaic, where the receptive field and corresponding dendritic fields of individual cells of the same subtype are arranged in territorial, regularly spaced grids[45,67,70]. Such arrangement leads to uniform coding of the visual scene across each retinal channel. This has remained unclear for ipRGCs, which overlap considerably, about 4-fold in the retina for the M1 ipRGCs[28]. As M1 ipRGCs participate primarily in non-image-forming vision, it is possible that spatial organization is de-emphasized in favor of maximizing the area of photon capture[71]. Our results show that M1 ipRGCs can be further subdivided and comprise independent subtypes that tile retinal space, like many of the well-described conventional bona fide RGC subtypes[24,70,72,73]. Indeed the coverage factor of SON-ipRGCs is similar to the average coverage factor of most RGC subtypes, identified from their functional receptive fields[68]. This indicates SON-ipRGCs are territorial and their dendrites overlap minimally. The functional significance of this arrangement is unclear, but it suggests, surprisingly, that retinotopy might also be important for circadian biology[74], and other non-image-forming functions mediated by the SON.

Our mapping data illustrates that the increased density of ipRGCs in the dorsal retina[3,43] arises from an additional population of ipRGCs found only in this region. There are two pieces of evidence supporting this conclusion; (1) when SON-ipRGCs are subtracted from retinal density maps, the dorso-ventral density gradient disappears and the number of M1 ipRGCs is equivalent in both hemispheres. (2) Retrograde labeling of SON-ipRGCs in *OPN4^Cre^* mice labels the same dorsal M1 ipRGCs as those localized in the *GlyT2^Cre^* mouse. These results

suggest that there are at least two independent populations of M1 ipRGCs in the dorsal retina, each with their own appropriate coverage factors. We estimate that SON-ipRGCs overlap approximately twice, whereas the total population of M1 ipRGCs in the dorsal retina overlap ~5.5-fold. These values were calculated using the average dendritic diameter of Neurobiotin-filled M1 ipRGCs in the $GlyT2^{Cre}$ mouse ($d = 324$ μm) and are slightly larger than the dendritic diameter of M1 ipRGCs previously described ($d = 274$ μm)[28], resulting in slightly different coverage values (Fig. S7c). Though these differences are likely due to labeling technique, we cannot exclude the possibility that SON-ipRGCs are larger than neighboring M1 non-SON-ipRGCs. Currently, it remains unclear if the ventral M1 ipRGCs belong to the same subtype of ipRGCs as those in the dorsal retina that overlap with the SON-ipRGCs. If the overlapping M1 ipRGCs follow the same spacing and distributions as the SON-ipRGCs, then there are at least two distinct subtypes of M1 ipRGCs in the rodent retina; one that is distributed across the entire retina and one that tiles the dorsal retina.

The dorsal-only location of SON-ipRGCs suggests retinotopy at the level of the dorso-ventral axis is fundamental for some types of non-image-forming vision. For the rodent, the horizon divides the visual scene into two distinct areas, consisting of differences in color, contrast, and behavioral relevance[40]. Accordingly, this is reflected in the asymmetric organization of cone photoreceptors across the dorso-ventral axis of rodent retina[40,41]. The higher density of UV-sensitive cones in the ventral retina enhances the dynamic range of photoreceptors for encoding the darker contrasts, which dominate the upper visual field and is likely important for predator detection[40,45]. Similarly, the dynamic range of the green cones that are in abundance in the dorsal retina is matched to encode the more even distribution of both light and dark contrasts found in the ventral visual field, likely aiding in navigation through foliage or burrows and finding small grains, grass and insects for consumption.

While rodent RGCs can form non-uniform topographic variations across the retina[75], SON-ipRGCs are a unique example of an RGC subtype restricted to a subregion of retina. Their location is almost identical to the region of the dorsal retina that is low in UV-sensitive cones, suggesting ipRGCs and cone photoreceptors may have adapted similarly to encode information that is asymmetrically distributed in visual space. Non-image-forming vision might also be adapted to encode the more uniform distribution of bright and dark contrasts, like cone photoreceptors. Alternatively, as the visual system of the rodent is optimized for nocturnal vision, they might require additional processing power in the night or daylight environment. Moon and starlight reflected off the ground is likely to predominate in nocturnal environments and reflected luminance might contain more useful information to drive the suppression of SON-mediated behaviors such as maternal activity or feeding[76,77].

The peptide PACAP is present in a dorsal population of ipRGCs in the rat[43] and both PACAP and the PACAP receptor $PAC_1$ are expressed in the SON[78,79]. Additionally, PACAP-positive retinal hypothalamic tract terminals are localized to an SCN region that resembles the outer-core projections of SON-ipRGCs. However, our own experiments to determine if the SON-ipRGCs and other $GlyT2^{Cre}$ expressing ipRGCs overlap with PACAP-expressing RGCs (Fig S4) failed to identify overlap, suggesting that PACAP-expression may be different in the mouse retina.

There are currently six known types of ipRGCs (M1–M6), primarily distinguished by their morphology[6,12]. However, recent evidence suggests additional functional subtypes likely exist within this current organization[10,15,17,21,35,69,80]. How do these additional ipRGC subtypes fit with our current understanding of M1 ipRGCs? Some ipRGCs do not express the transcription factor Brn3b and this small number of ipRGCs projects to the SCN and IGL only, avoiding other brain regions[17]. Our results suggest that a separate tiling subpopulation of M1 ipRGCs co-innervates the SON, the outer core of the SCN, and the IGL, likely influencing distinct behavioral functions.

The SON is a collection of secretory cells that participate in the hypothalamic-pituitary-adrenal axis by producing antidiuretic hormone (ADH) and oxytocin. ADH is responsible for regulating water reabsorption in the kidneys[81] and oxytocin plays a critical role in lactation and parturition[82]. The significance of visual input to this area via the SON-ipRGCs is unclear but circadian changes in urine volume and concentration[83,84], as well as patterns of lactation[85,86] are well established in humans and animal models. Like the influence of ipRGCs on entrainment of the SCN, bright light might act as a Zeitgeber in the SON, keeping the daily release of AVP well-timed or to aid in adjusting fluid balance to altered light cycles, such as when changing time zones. Alternatively, ipRGCs that innervate the SON might be involved in more direct effects of light on the release of AVP or oxytocin by SON neurons. It is also possible that SON-ipRGCs regulates the release of oxytocin, AVP, or other neuropeptides such as cholecystokinin, or CART, throughout the brain, rather than in the pituitary, where SON-ipRGCs might be important for regulating direct light-activated influence on maternal behaviors, feeding[87], or social behaviors[56,57].

The unique innervation of SON-ipRGCs to known circadian structures (SCN and IGL) is also of significant interest. Oscillatory activity in the SCN functions as a circadian timing circuit, predicting physiological and behavioral needs throughout the day and night[88]. The SCN is divided into two distinct subdivisions, designated as core and shell, based primarily on localized peptidergic expression, innervation, and projection[89]. The core is the site of direct visual input from ipRGCs, indirect visual input from IGL, and is localized by the expression of a vasoactive intestinal polypeptide (VIP) and gastrin-releasing peptide (GRP)[46]. The SCN shell contains a large number of AVP neurons and receives innervation from other CNS nuclei[46,90]. Our results show that SON-ipRGCs innervate a localized region of the SCN we call the outer core, avoiding both the central core of primary ipRGC input and the shell delineated by AVP neurons. As SON-ipRGCs represent a distinct subtype of ipRGC it suggests that the SCN receives at least two types of direct retinal input, segregated to at least two localized areas of the SCN. The behavioral relevance of this organization is unclear, but given the localized distribution of SON-ipRGCs in the retina, and their overlap with other M1 ipRGCs, it suggests that subtypes may be encoding different aspects of environmental light. Given their broad projections to the SCN, IGL, SON, and the limited knowledge of their specific connectivity within the SON, behavioral analysis of the specific functional role of this unique subtype would require surveying many behaviors that might rely on subtle inputs from SON-ipRGCs.

Why are there ipRGCs labeled in the $GlyT2^{Cre}$ line, which, other than ipRGCs, labels predominantly glycinergic amacrine cells in the retina? This question is particularly prescient given the recent discovery of GABAergic ipRGCs that project to the SCN, IGL, and OPN[69]. Our results, however, indicate that SON-ipRGCs do not release GABA or glycine, and, thus, must form a separate population from GABAergic ipRGCs. It remains unclear if they express the $GlyT2$ transporter in their axon terminals, and if they do, what functional role the transporter might play in modulating central synapses. It is possible that SON-ipRGCs release glycine to modulate the glycine binding site on post-synaptic NMDA receptors. Alternatively, their labeling in $GlyT2^{Cre}$ mice may be some other function of the bacterial artificial chromosome (BAC) insertion in this particular transgenic line similar to other transgenic lines like the $HB9^{GFP}$ line that labels ON-OFF direction-selective ganglion cells due to the genomic insertion site of the BAC[91]. Regardless, our anatomical mapping data predicts there are likely only two subtypes of ipRGCs in the dorsal retina, if other ipRGCs follow similar mosaic spacing rules as SON-ipRGCs. While we do not know if GABAergic ipRGCs are M1 ipRGCs, their projections to the SCN strongly suggest some of them are, whereas their projections to the shell of the LGN might indicate some are not ipRGCs[69]. Given GABAergic ipRGCs are more numerous in the dorsal retina, it remains

unclear if they are the other subtype we predict to lie in the dorsal retina or, instead, they might be part of multiple subtypes of ipRGCs that do not form complete mosaics throughout the retina. Future studies of both retrograde tracing of these populations and functional recording from ipRGCs in the retina and brain are required to resolve these questions.

## Methods

Experiments involving animals were in accordance with the National Institutes of Health guidelines, and the Oregon Health and Science University Institutional Animal Care and Use Committee approved all procedures (protocol TR02IP00000096). Animals of both sexes were used in this study and data from both sexes were combined. Due to the low number of animals, we did not perform a sex-specific analysis. Where sex was recorded, the sex of the mouse is listed in the source data file. The overall number of mice where sex was recorded in this study was 25 females and 29 males. Mice were housed at an ambient temperature of 20−21 °C, and humidity of 40–60% on a 12:12 h light:- dark (LD) cycle, with free access to food and water. *GlyT2Cre* mice (Tg(Slc6a5-cre)KF109gsat/Mmucd) were a gift from Larry Trussell, prior to being cryo-recovered by the OHSU Transgenic Mouse Model Core using sperm purchased from the Mutant Mouse Resource and Research Center (Stock 030730-UCD). Ai32 ((B6.Cg-Gt(ROSA) 26Sortm32(CAG-COP4*H134R/EYFP)Hze/J) strain 024109), Ai9 (B6.Cg- Gt(ROSA)26Sortm9(CAG-tdTomato)Hze/J; strain 007909), PACAP-2A- IRES-Cre (B6.Cg-Adcyap1tm1.1(cre)Hze/ZakJ; strain 030155), and *Ai140* (Ai140(TIT2L-GFP-ICL-tTA2)-D; strain 030220) mice were obtained from The Jackson Laboratories. *OPN4Cre* (tm1.1(cre)Saha/J) were a gift from Samer Hattar and Johns Hopkins University. Animals were bred and housed on a 12-h light/dark cycle with food and water ad libitum.

To trace ipRGC projections and sites of central innervation, anterograde tracers were delivered to the eye through intravitreal injection. *AAV-FLEX-tdTomato* (Cat# 28306 AAV2 & PHPeB), *AAV1-DF- ChR2-mCherry* (Cat..# 18916), and AAV *pCAG-FLEX-EGFP-WPRE* (Cat.# 51502) *AAV-CAG-DIO-ChR2(H134R)-EYFP* (Cat #127090; AAV PHP.Eb) were purchased through Addgene. For this procedure, animals were anesthetized by intraperitoneal injection of 100 mg/kg ketamine and 15 mg/kg xylazine. Proparacaine and tropicamide drops were applied topically to the eye for local anesthesia and to improve visualization of the surgical field, respectively. Under stereo microscopic control, the ora serata was incised 32 G needle. AAV vectors were delivered in 1.5 μL volumes to the vitreous of the eye using a 5 μL Hamilton microinjection syringe or a Drummond Nanoject 2. To aid in visualizing retino- recipient brain structures, animals also received a secondary eye injection of 1 uL CTB-488 7d before sacrifice. In order to identify ipRGCs that innervate specific central locations, stereotactic brain injections of retrograde tracers were performed using a Kopf stereo- tactic instrument. About 92 nL of *AAVRG-DF-ChR2-mCherry* (Catalog# 18916) was injected bilaterally at 0.5 mm from Bregma, +1.3 mm lateral to the midline at a depth of 5.0 mm, determined from the Paxinos & Franklin Mouse Brain Coordinate Atlas, 5th ed[92]. Animals were sacri- ficed for 3 to 4 weeks following injections for brain and eye histology.

For retinal histology, animals were euthanized with 200 mg/kg ketamine and 30 mg/kg xylazine, followed by cervical dislocation. Eyes were then removed with curved surgical scissors and placed in 4% paraformaldehyde (PFA; Electron Microscopy Sciences Catalog#: IC993M31) + PBS for 30 min with the cornea partially removed. Eyes were then washed thoroughly in PBS for 24 h. Dorsal-ventral orienta- tion, marked with a ventral cut, was established using the choroid fissures and retinal artery, which can be visualized entering the caudal portion of the sclera, inferior to the optic nerve. The whole retina was then transferred to a 1.5 ml Eppendorf tube for immunohistochem- istry. The details, including the timing and concentration of primary and secondary antibodies used for specific experiments, are described in Supplementary Table 1. Retinas were dried on the slide until

transparent, then mounted with a coverslip using Vectashield mounting medium.

We used the following antibodies: Rabbit anti-melanopsin (ATS- Bio - Cat#: AB-N39); Chicken anti-mCherry antibody (Abeam - Cat#: ab205402); Goat anti-GFP antibody (Abeam - CAT#: ab5450) Chicken anti-GFP antibody (Aves Labs GFP-1010); Guineapig anti-RBPMS (Phosphosolutions 1832-RBPMS); Goat anti-cholera toxin subunit B (List Labs - Cat#: 703); Rabbit anti-vasopressin antibody (Immunostar - Cat#: 20069); Rabbit anti-opsin antibody blue (Millipore - Cat#: ab5407); Mouse anti-Neurofilament H (SMI-32 Biolegend; previously Covance Cat SMI-32); Donkey anti-rabbit Alexa Flour 647 (Jackson lmmuno Research - Cat#: 711-605-152); Donkey anti-goat Alexa Flour 488 (Jackson Immuno Research - Cat#: 705-545-147); Donkey anti-chicken Alexa Fluor Cy3 (Jackson lmmuno Research - Cat#: 703- 165-155).

Rabbit anti-melanopsin (Advanced targeting systems cat AB-n39) is an affinity-purified version of the n-38 antibody (RRID: AB_1608077) and has been previously published[93], which states that it "is raised against a synthetic peptide consisting of the 15 N-terminal amino acids of mouse melanopsin (MDSPSGPRVLSSLTQ). It produces no staining in mela- nopsin knockout mice (Opn4$^{-/-}$)." Using the OPN4Cre mouse line, we also validated this antibody produces no staining in OPN4Cre$^{+/+}$ mice, which are OPN4$^{-/-}$ or knockout mice[17]. Chicken anti-mCherry antibody (Abcam ab205402; RRID AB2722769) is raised against the recombinant protein (His-tag) corresponding to mCherry (sequence from ref. [94]). This antibody stains a band with the predicted molecular weight of 30 kDa in a lysate of HEK293 cells transfected with pFin-EF1-mCherry vector. Goat anti-GFP antibody (ab5450; RRID AB_304897) is a poly- clonal antibody raised against the recombinant full-length protein corresponding to GFP. This antibody produces signal amplification in cells expressing GFP. Chicken anti-GFP (Aves labs GFP-1020; RRID AB_2307313) is a polyclonal antibody raised against recombinant GFP expressed in *Escherichia coli*. The vendor states, "Antibodies were ana- lyzed by western blot analysis (1:5000 dilution) and immunohis- tochemistry (1:500 dilution) using transgenic mice expressing the GFP gene product. Western blots were performed using BlokHen® (Aves Labs) as the blocking reagent, and HRP-labeled goat anti-chicken anti- bodies (Aves Labs, Cat. #H-1004) as the detection reagent". In our experiments, this antibody strongly amplifies the signal that is pro- duced in mice where Cre drives the production of EGFP in specific neuronal populations (GlyT2Cre;Ai140; Ai140 from Jackson mouse line 030220). Goat anti-cholera toxin subunit B (List Labs - Cat#: 703; RRID AB_10013220) is a goat polyclonal antibody against the Cholera Toxin B subunit. In our experiments, this antibody produces strong amplifica- tion of the axon terminals of retinal ganglion cells in the suprachias- matic nucleus following eye injections of Cholera Toxin B subunit, as previously published in other papers using the same methods[4] and when conjugated to a secondary antibody with a far-red (Alexa 647) fluorophore. Rabbit anti-vasopressin antibody (lmmunostar - Cat#: 20069; RRID AB_572219) is a rabbit polyclonal antibody against arginine vasopressin. The vendor states, "The antibody produces significant fluorescent staining and significant biotin-avidin/HRP staining at a 1/2000–1/4000 dilution in rat hypothalamus. Staining is completely eliminated by pretreatment of the diluted antibody with 10 μg/mL of arginine vasopressin. Preadsorption with as much as 100 ug/mL of oxytocin had no effect on immunolabelling". In our experiments, this antibody labeled neurons in the suprachiasmatic nucleus, the para- ventricular nucleus, and the supraoptic nucleus as published in pre- vious studies (refs. [95,96]). Rabbit anti-opsin antibody blue (Millipore Sigma - Cat#: ab5407; RRID AB_177457) is a rabbit polyclonal antibody against recombinant human blue opsin. Validation information can be found on the manufacturer's website: https://www.emdmillipore.com/ US/en/product/Anti-Opsin-Antibody-blue,MM_NF-AB5407.    In    our experiments in mice, this antibody-stained S-cone photoreceptor outer segments located predominantly in the ventral retina, as previously

published (refs. [97,98]). Mouse anti-Neurofilament H (SMI-32 Biolegend; previously Covance Cat SMI-32;RRID AB_509997) is a mouse affinity-purified monoclonal antibody against neurofilament H. Validation information can be found at the manufacturer's website: https://www.biolegend.com/en-us/products/purified-anti-neurofilament-h-nf-h-non phosphorylated-antibody-11475. In our experiments SMI-32 labeled alpha-type retinal ganglion cells as previously published (refs. [75,99]). Guinea Pig anti-RBPMS (Phosphosolutions Cat# 1832-RBPMS lot NB322g) is a polyclonal antibody raised against a synthetic peptide corresponding to amino acid residues from the N-terminal region of rat RBPMS, conjugated to keyhole limpet hemocyanin (KLH). This antibody is validated by Pérez de Sevilla Müller and colleagues[100] and, in our experiments, labels retinal ganglion cells but not retinal amacrine cells.

For brain histology, animals were heavily anesthetized by IP injection of ketamine/xylazine and transcardially perfused with 50 μL Heparin + 30 mL PBS followed by 40 mL 4% PFA + PBS. Brains were post-fixed in 4% PFA + PBS for 2–4 h and then washed thoroughly in PBS for 24 h, mounted in 4% agarose, and vibratome sectioned at 200 μm. Sections were collected in PBS and transferred to glass slides. Retinas and brain slices were immunostained in a mixture of 5% Donkey serum, 2% bovine serum albumin, 0.5% Triton-X-100, and 0.025 % sodium azide at room temperature. Details, including the timing and concentration of primary and secondary antibodies used for specific experiments, are described in Supplementary Table 1. Both brain slice and whole-mount retina were mounted using Vectashield mounting medium (Vector Laboratories) and imaged on a Leica SP8 scanning confocal microscope.

To generate retina maps, whole retina tiling confocal z-stacks were captured using a 40x oil objective. Tiles were stitched together in Leica LAS X Life Sciences software and analyzed in ImageJ. ipRGCs were manually identified across the entire retina by systematically localizing all melanopsin-positive cell bodies in 200 × 200 μm square increments. Somas were marked as regions of interest (ROI) in a separate overlay image using the multipoint tool in ImageJ (2d axis image). M1 ipRGCs were identified by their characteristic dendritic stratification in sublamina-a of the IPL, their small somas, and bright melanopsin staining. $GlyT2^{Cre}$ ipRGCs were identified by the co-localization of EGFP and melanopsin in their cell bodies ($GlyT2^{Cre};Ai140$). All $GlyT2^{Cre}$ RGCs were identified by co-localization of RPBMS and TdTomato in their cell bodies ($GlyT2^{Cre};Ai9$), then determining the fraction that also expressed melanopsin (Fig. 3i–k). The sub-portion of $GlyT2^{Cre}$ that were M4 ipRGCs we identified by co-localization of SMI-32[101], melanopsin, and TdTomato in their cell bodies (Fig. S3). Melanopsin distribution maps of ipRGCs were generated from the X/Y coordinates extracted from the axis image. Due to their abundance, UV cone distribution maps were generated using the trainable Weka segmentation plugin for ImageJ[102] (imagej.net/plugins/tws/). This machine-learning software allows structures of similar appearance to be identified in a semi-automated manner. Images were processed with a binary threshold and segmentation was trained to identify fluorescent cells (UV + cone outer segments) in the photoreceptor layer. Segmentation can be challenging when the proximity of cells is small or overlapping. As a result, the density of UV cones reported in the ventral retina is likely underestimated.

Neighbor density maps and density recovery profiles were generated using the Neighbor density analysis application within the BioVoxel_Toolbox plugin for ImageJ (imagej.net/plugins/biovoxel-toolbox). Axis images denoting cell bodies were converted to 8-bit and applied with a binary threshold. Particle neighbor analysis was used to identify the number of cell bodies within a given radius from each soma. To compare with a random distribution, we modeled a randomized organization of points using an ImageJ macro (https://imagej.nih.gov/ij/macros/DrawRandomDots.txt) matched for density and number of cells. We then performed identical particle neighbor analysis on these data sets as described above. Neighbor density maps were generated for SON retro-labeled cells and the modeled random distribution of cells with a density radius of 220 μm to approximate the average dendritic diameter of RGCs. Density recovery profiles were calculated similarly using radii from 0 to 400 μm in 20 μm increments. Nearest neighbor (NN) distance was determined for SON retro-labeled cells and the modeled random distribution of cells using the Nearest Neighbor Distance plugin for ImageJ. (https://icme.hpc.msstate.edu/mediawiki/index.php/Nearest_Neighbor_Distances_Calculation_with_ImageJ.html). Regularity index was calculated by computing the ratio of the mean nearest neighbor distance to its standard deviation per retina. Nearest neighbor regularity indices are used to measure the degree of regularity in the distribution of cells. Population per hemisphere was calculated by dividing oriented retinas through the optic nerve head along the naso-temporal axis and quantifying the number of cells per hemisphere. Cell density was determined by quantifying the number of cells within 1 mm² areas of the dorsal and ventral retina.

The central innervation pattern of ipRGCs was examined in $GlyT2^{Cre}$ mice following AAV eye injection using $AAV2-FLEx-EGFP$ and in $GlyT2^{Cre}$ and $OPN4^{cre}$ mice following SON retrograde brain injections using $AAVRG-DF-ChR2-mCherry$. To improve repeatability and consistency the eye injections and brain injections were performed using a Kopf stereotactic instrument. Brain sections (200 um) were then stained and imaged under confocal microscopy. Confocal images were then projected into two-dimensional images. Brain regions were localized using DAPI staining, immuno-staining of arginine vasopressin, and referenced to the Paxinos & Franklin Mouse Brain Coordinate Atlas[92]. Average fluorescence values were measured by generating a mask around the pertinent brain structures using ImageJ's selection tools (See regions denoted by dotted lines in Fig. 5 and Fig. S10). Values were then normalized to background fluorescence values within the brain slice (F/Fo) and graphed in GraphPad Prism (version 9). Values >1 represent the average fluorescence of identified brain structures above the background. Regions, where structures could not be confidently identified, were not included in the quantification. Cross-sectional fluorescence through brain structures were used to compare patterns of innervation across a particular brain region. This was performed by generating a mask of 30 um × 750 um (SCN) or 30 um × 650 um (vLGN, IGL, DLG, and OPN) across the same confocal images, transecting structures as illustrated in Fig. 5 and Figs. S10, S14, S15, S16). Measurements of average fluorescence values per 0.75 um distance were then collected in ImageJ, binned at 6 um, averaged across brains, and plotted as profiles depicting the average fluorescence values through coordinal planes of the SCN, vLGN, IGL, DLG, and OPN for eye injected and brain injected animals.

Single-cell current-clamp recordings were performed in the $GlyT2^{Cre};Ai9$ 9 mouse using a HEKA EPC800 amplifier, ITC-18 digitizer, and Axograph software (version 1.7.4) or Patchmaster software (v2x73.5 and v2x90.5 (HEKA Elektronik Dr.Schulze GmbH). Fluorescent ipRGCs were targeted using brief 554 nm exposure (<100 ms) and a high-sensitivity camera (Andor technologies−DU-888E-COO-#BV). Recordings were performed in Ames medium with synaptic blockers 20 μM L-AP4, 25 μM DAP5, 20 μM CNQX to isolate ipRGCs. Five-second illumination of blue (445 nm) light was used to elicit intrinsic melanopsin responses at $5 \times 10^{13}$ log photons/cm$^{-2}$/s$^{-1}$ using a Texas Instruments DLP4500 LightCrafter projector and custom software (pyStim[103]; https://github.com/SivyerLab/pyStim). Some ipRGCs were targeted for Neurobiotin electroporation using methods described previously[104,105].

To study synaptic transmission mediated by the axons of SON-ipRGCs, $GlyT2^{Cre}$ mice were crossed with the Ai32 reporter mouse driving ChR2 expression in $Cre$ expressing ipRGCs. Male and female $GlyT2^{Cre};Ai32$ mice were housed in an environmental chamber (Percival Scientific, Perry, IA) maintained at 20−21 °C on a 12:12 h light:dark (LD) cycle, with free access to food and water. The ChR2 expressing axonal terminals projecting to the SCN and SON were activated by white light passing through a Chroma excitation filter (BP 470/40). The estimated

intensity of the light was 16.5 to 17 log photons/cm$^{-2}$/s$^{-1}$. The ChR2 expressing RHT projections were observed using a YFP filter (Chroma, ET-EYFP C212572, Cat.# 49003). The recordings were performed at the end of the day and the beginning of the night. SCN and SON neurons were voltage-clamped in the whole-cell and cell-attached patch-clamp modes. The cells were filled with Neurobiotin (0.5%), which made it possible to determine their localization after the experiment. The internal solution consisted of (in mM): 87 CH$_3$O$_3$SCs, 15 CsCl, 1 CaCl$_2$, 10 HEPES, 11 EGTA, 31.5 CsOH, 3 MgATP, 0.3 TrisGTP, 10 Phospho-creatine di(tris) salt, and 5 N-(2,6-dimethylphenylcarbamoylmethyl) triethylammonium chloride (QX-314); pH 7.25, 278 mOsm. The artificial cerebrospinal fluid (ACSF) was (in mM): 132.5 NaCl, 2.5 KCl, 1.2 NaH2PO4, 2.4 CaCl2, 1.2 MgCl2, 11 glucose, and 22 NaHCO3, saturated with 95% O$_2$ and 5% CO$_2$; pH 7.3–7.4, 300–305 mOsm. The equilibrium potential for chloride was −50 mV. For extended detail, see refs. [106,107]. During recordings the inhibitors of glycine, GABA$_A$ receptors, and ionotropic glutamate receptors, respectively strychnine (1 μM), gaba-zine (10 μM), CNQX (20 μM) + AP-5 (DL-AP-5, 50 μM), and TTX (1 μM) an inhibitor of voltage-dependent Na$^+$ currents were applied. EPSCs evoked by electric stimulation of the optic chiasm were also used to confirm the recorded cell received retinal inputs. Analysis of electrophysiology data were performed in Axograph (version 1.7.4) and Igor Pro (Wavemetrics; version 6.22 A, 6.37, 7.08, and 8.04). Statistical analysis was performed in GraphPad Prism (version 9).

## Statistics and reproducibility

Statistical analysis is indicated in the figure legends or in supplementary tables where applicable. For data where single or representative micrographs are shown, we repeated each experiment independently with similar results as follows: Figs. 1a, e = 14 animals; 1c, d = 7 animals; 3a, b = 14 animals; 3c = 1 animal; 4b, c = 21 animals; 5a, b = 4 animals; 5c = 1 animal; 5f = 4 animals; 5i = 4 animals; 5j = 6 animals; 5m = 4 animals; 6c, f = 3 animals; 7c, d = 3 animals; 8c = 11 animals; 8n = 6 animals; 9b, c = 7 animals. Supplementary Figs. 1a, b = 14 animals; 3a = 3 animals; 7a–d = 1 animal; 8a, b = 4 animals; 9a-l = 5 animals; 10a = 5 animals; 14a, c = 6 animals; 14b, d = 3 animals; 15a = 6 animals; 15b = 3 animals; 16a = 6 animals; 16b = 3 animals; 18b = 13 animals.

## Reporting summary

Further information on research design is available in the Nature Portfolio Reporting Summary linked to this article.

## Data availability

All data generated or analysed during this study are included in this published article (and its supplementary information files). The processed electrophysiology data and raw traces are provided in the Supplementary Information/Source Data file. Raw confocal microscopy data will be provided upon reasonable request. Source data are provided with this paper.

## Code availability

Visual stimulus software (pyStim) is freely available on Github: https://github.com/SivyerLab/pyStim.

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

## Acknowledgements

We would like to thank Alex Tomlinson for help with pyStim, Andre Dagostin's help with mouse husbandry and pilot experiments, Lane Brown for help obtaining *OPN4^Cre^* mice, and David Vaney for critically reading the manuscript. We would like to thank Benjamin Reese for their critical comments on the implications of the retinal mosaic analysis. This work was supported by NEI EY032564, EY027202, Lloyd Research Fund, Medical Research Fund of Oregon New Investigator grant, P30 EY010572, and unrestricted departmental funding from Research to Prevent Blindness (New York, NY) to B.S., DC004274 to H.v.G., NS103842 to C.A.N., EY031984 to M.H.B., EY032057 to K.M.W., and acknowledgement is made to the donors of ARCs scholarship program, and the National Glaucoma Research, a program of BrightFocus Foundation, for support of this research.

## Author contributions

M.H.B., M.M., M.A.M., T.G., O.C., E.W., C.N.A., J.L., H.v.G., and B.S. designed and performed experiments and analyzed data, with input

from K.M.W. Confocal microscopy and analysis was performed by M.H.B., J.L., and B.S., retina patch-clamp recordings and dye filling and eye injections were performed by M.H.B., brain injections were performed by O.C., brain slice recordings and optogenetics were performed by M.M., M.A.M., and T.G. with help from M.H.B. The manuscript was written by M.H.B. and B.S. with input from all authors.

## Competing interests

The authors declare no competing interests.
