## [Peer Review File · Nature Communications]

A melanopsin ganglion cell subtype forms a dorsal retinal mosaic projecting to the supraoptic nucleusREVIEWER COMMENTS

Reviewer #1 (Remarks to the Author):

This manuscript from Barrie et al. is a very nicely done characterization of ipRGCs that are labeled in the GlyT2 Cre line. The authors describe the subtypes, projection patterns, spacing and neurotransmitter content of these ipRGCs, finding that they generate an independent mosaic of ipRGCs in the dorsal retina overlaid on the whole retina ipRGC mosaic. Overall, this work will be a welcome addition to the field. My concerns are addressable and outlined below.

1. The quantifications and images of the density/coverage/etc. and the anatomy are both quantitative and visually striking.
2. The title and abstract make it sound as though the paper is going to describe how this new mosaic of ipRGC encodes ground luminance. Beyond describing the fact that this mosaic exists in the dorsal retina, this question of how it encodes luminance and how that information is used is not further pursued, which was jarring as a reader. The title and abstract should be adjusted to more accurately portray the content of the paper, which is to characterize the labeled RGCs in this mouse line. The idea that the dorsal retina gets input from the ventral visual field is not, for example, a new idea.
3. The authors state that this population primarily consists of M1 ipRGCs, but 16 of the 27 filled RGCs are ON stratifying, i.e., not M1 ipRGCs. The authors do not provide other quantification to support that this is majority M1, which they could do with their retinal density measurements since they are able to identify M1 ipRGCs using immunolabeling. The overall proportion of GlyT2 ipRGCs and SON-ipRGCs that are M1 vs. not should be carefully quantified. The SON ipRGCs were only loosely quantified as M1 (~97% but unclear where that number came from). It would be helpful to show confocal images supporting that the authors can truly identify stratification of SON-projecting cells (I don't doubt that they can, but they should demonstrate it). Moreover, the small ON cells could be temporal M4 cells, have the authors looked at SMI32 labeling?
4. The authors acknowledge that some non-dorsal ipRGCs are labeled in their retrograde injections. They give good reasons that this is the case, but still the conclusion that dorsal ipRGCs are the sole input to the SON has not been conclusively demonstrated until spillover has been ruled out. The authors should soften this language in line 237, it does not detract from the findings to do so.
5. The authors briefly characterize the morphological features of SON M1 ipRGCs, but it would be useful to compare this to other dorsal M1 ipRGCs to determine if there are distinguishing morphological features of this population, this would be an important addition to our emerging understanding of M1 variability/diversity.
6. The authors do not mention whether they saw SON-ipRGC collaterals in the shell of the OPN or other targets. They should note whether each of the targets that receive GlyT2 ipRGC axons receive SON-ipRGC collaterals, this will be important in identifying 1. Potential projections of the non-M1 GlyT2 ipRGCs and 2. Give a more comprehensive overview of the innervation patterns and types of GlyT2 ipRGC inputs to each area.
7. It would be useful to know, since the authors address neurotransmitter content of these SON-ipRGCs, whether they actually express GlyT2 mRNA or protein. This will provide important insights into the biology of these cells (i.e. are capable of packaging glycine) and define whether this labeling is simply a feature of the BAC line, as they suggest, but does not represent actual gene expression. This is particularly useful if others may want to use this as a marker of this ipRGC population in future studies.

Reviewer #2 (Remarks to the Author):

In this interesting and thorough study, the authors have identified a new subtype of ipRGC. By exploring the expression pattern of the retina of the GlyTr-Cre BAC transgenic mouse, the authors found that in addition to labeling inhibitory interneurons, a subset of dorsally located retinal ganglion cells were also labeled (RGCs are identified by the presence of an axon). They show that some of these RGCs as ipRGCs based on 1) co-staining with melanopsin 2) their projection targets and 3)

their strong intrinsic light response. Intracellular fills show that they are a mixed population – some with dendrites stratifying in the ON sublamina and some stratifying in the OFF sublamina, the latter property consistent with them being M1-ipRGCs.

The authors then focus the rest of the study in these dorsally located OFF-stratifying M1 ipRGCs, focusing on whether they form a unique subclass from the other M1 ipRGCs with which they are intermixed. One piece of evidence is their “mosaic tiling” (more on this below). They also show they have central projection targets distinct from other M1s - namely the “outer core” of the SCN (between the shell and the central core) and that they also innervate the SON. Finally, in a set of heroic experiments, they express channelrhodopsin in all of the GlyTr-Cre RGCs and show they form glutamatergic synapses with targets in the IGL and SCN. The authors also propose a unique function for this cell type – its dorsal location indicating they are uniquely tracking luminance on the ground, though the physiological significance of this is speculative.

I have few criticisms of the technical aspects of the project – it is all extremely well done. This work represents a significant contribution to our understanding of the retina and its influence on physiology. One general comment is that the authors have used a “kitchen sink” approach for each of the submethods – which sometimes crowds out the experiments that are more definitive and leaves the reader more confused than necessary. Hence the authors are encouraged to streamline when possible.

Below are some specific comments regarding aspects of the study that need to be clarified.

1. The authors emphasize that the presence of “a regularly spaced mosaic” (e.g. line 88, line 243) and argue this organization is indicative of correct identification of a cell type. This mosaic argument is based on the presence of an exclusion zone in the density recovery profile - DRP (e.g. SON-ipRGCs form an ordered mosaic on the dorsal retina). The authors are strongly encouraged to read recent review Keeley 2020 (and Reese 2015) that make the case that the presence of an exclusion zone is not conclusive evidence that a retinal mosaic is regular and significantly different from random, especially with sparsely distributed cells such as SON-ipRGCs. One alternative is to compare the measured DRP with the DRP computed from a random distribution of cells matched for soma size and density. That said, I don't think the lack of “regular” mosaic will significantly weaken their results – but the authors are cautioned on relying too much on this method for defining a cell type and should remove reference to the soma organization being “regular” throughout. (I would have classified this as a “minor” point except that the authors have emphasized this analysis throughout.)
2. As is clearly the case here, BAC-transgenic in general cannot be used to cleanly identify cell types. Indeed the multiple other types of cells labeled in this line – in particular the non-M1 ipRGCs, can complicate the interpretation of several experiments. The authors need to quantify what percentage of labeled RGCs in the GlyT2-cre mouse line are non-ipRGCs (line 116). For example, in Figure 1, they do not give the percentage of RGCs expressing eGFP (or td tomato in Figure S1) that are melanopsin positive nor do they give a quantification of percent of cells they patched that have a light response.
3. As noted by the authors, a subpopulation of dorsal ipRGCs were previously described as expressing PACAP (Hannibal et al 2002) (line 478). This point feels rather buried in the discussion when there have been several studies regarding the role of PACAP in the SON and its influence on other circadian functions. Rather than a side point, this is important context for the current study and should be brought up in the intro. Perhaps more importantly, it seems straightforward for the authors to test whether these cells are the same subpopulation - namely, are the SON-ipRGCs PACAP positive? Knowing whether these cells are PACAP positive would also provide a potential way to genetically identify these ipRGCs, as so far there have been no specific marker genes identified for SON-ipRGCs.
4. Similarly, the analysis of the various projection targets is quite detailed and impressive but seems critical to know if non-ipRGCs or ON-stratifying ipRGCs comprise some of these targets, particularly in areas outside of exclusive ipRGC target regions such as zona incerta (e.g supplementary Figure 5).

5. Figure 8 and the supporting text is a bit hard to figure out. First, the figure caption doesn't match the lettering on the figure. Second, line 335 mentions mapping locations of postsynaptic SON neurons by matching to a DIC image but this is not in the figure. It is also not clear what is measured in a "cell attached voltage clamp mode" – this is just an unclamped AP, correct? The value of these measurements is not clear particularly when there are whole cell voltage clamp experiments as well. Also the motivation for using the Opn4-Cre in Part O is not clear for these purposes. Finally, no quantification is given for Parts N and O so the conclusions for these coronal sections seem rather limited and the authors should consider removing.

6. Why are the authors including Figure 9a-g? This seems like a poor experiment since they ultimately decide that there are multiple sources of input to the IGL neurons and therefore not that interpretable regarding retinal inputs. I would recommend eliminating this part of the figure and stick to the more interpretable component regarding Figure 9h-n. If there is some other motivation for 9a-g, then the authors need to provide it. And if 9a-g is retained, the summary data in d-e should make clear which data was measured in the presence of single synaptic blockers vs combinations.

7. Supplementary figure 7d - remove the brown points indicating "predictive SON projecting," since that unreasonably assumes a priori that the mosaic spacing of SON-ipRGCs in the shown area is regular.

8. Lines 282/supplementary Figure 8 – it was unclear the motivation behind looking at these projections or exactly what is being shown here.

9. Minor point: In Figure 2, an analysis of dendrites is used to make the case that OFF-stratifying labeled cells are similar morphology to M1s (line 133) – however no reference is given for what is known about M1 dendritic morphology and how strong this mapping is. The authors may at least want to add the reference or state explicitly that the morphology is just similar at a gross assessment.

Reviewer #3 (Remarks to the Author):

In this manuscript, the authors report a subtype of ipRGCs that form a regularly spaced mosaic limited to a dorsal region of the retina, selectively projecting to the supraoptic nucleus. Using a combination of confocal microscopy, anterograde and retrograde labelling, patch clamp recordings and optogenetics, the authors show that ipRGCs can be further subdivided and comprise independent subtypes that tile retinal space, like other conventional RGCs reported previously, and suggest that mice devote additional melanopsin-dependent processing power to their ventral visual field. I find this manuscript provides interesting findings and a perspective on a possible role of the ipRGCs and their specific projections to the lateral hypothalamic brain regions. I only have minor comments, which largely are requests for additional analysis details and further clarifications of their findings for better presentation.

- Lines 100-121 (Fig1 and S1)

I wonder whether all the analysis regarding the projections of RGC axon terminals in the brain were done manually and visually. Conventional manual processing is highly subjective and sample-specific, and even precise comparison across sample animals is impossible. Thus, can you show any quantitative data or analysis to confirm your conclusion of systematic mapping? – If possible, it will be helpful to show some numbers and statistics such as the number of cells measured in each RGC and the average portion of the projection in the target area, etc. in each animal. Another possible scenario is to use some pre-published algorithms and software (For example, see Song and Choi et al. Cell Reports 2020) to compare datasets from different mice successfully on a properly calibrated reference space. In any cases, it would be good to provide detailed numbers and statistics so that other researchers can reproduce your findings and compare with.

- Lines 135-145 (Fig 3)

This is one of the main results of this study, but it seems that only a single animal case is shown. How many different animals were used to confirm this result, and how the result varies across animals? Particularly, on the right graph of Fig 3d-f (density at each retinal location along the dorso-ventral axis), you can show the result in each animal separately and then plot the averaged one.

- Lines 174-187 (Fig 5)

The results in Fig 5 are mostly described qualitative using images only, which requires a subjective visual inspection. Probably, the bottom line of the result that authors claimed might be acceptable as it is, but it would be very helpful to try some quantitative analysis or plots of graphs that describes everything in more precise form. In addition, for details, you can describe how did you define each brain region, count the number of projections in each area, and how robust your observation across different animals, etc.

- Lines 253-266 (Fig 7 and S7)

Here it seems important to describe the mosaics and corresponding analysis together, so why don't you include materials in Supplementary Figure 7d in the main figure? Instead of the current panels Fig. 7a and b that are too small to see some details of mosaics structure, probably you can show an illustration of cell mosaics as that of the RGC receptive fields previously reported (For example, see Fig 1. in Anishchenko et al. Journal of Neurophysiology 2010), using the cell density and the coverage factor estimated here. This information is important enough to be in the main manuscript, and may help readers to easily examine a spatial organization of the population.

- Lines 402-404

"when SON-ipRGCs are subtracted from retinal density maps, the dorso-ventral density gradient disappears and the number of M1 ipRGCs is equivalent in both hemispheres."

: This seems quite an interesting observation, but I can't find a relevant plot for this.

- Minors

Lines 170-178.

Figures need to be rearranged. Fig S5 and S6 appear before S3 in the main text.

Fig 3. Some panels are missing or mislabeled. Panels h and i should be g and h.

Fig. 8. Panels 8c and d are too small to get any information visually. Can you add an inset for a "zoomed-in" local shot for better visibility?

A melanopsin ganglion cell subtype forms a retinal mosaic projecting to the supraoptic nucleus

Response to reviewers

We would like to thank all of the reviewers who gave positive, highly constructive, and helpful feedback. We have made major revisions and clarifications and in doing so have performed a number of additional experiments and analyses. The manuscript now includes 11 additional supplemental figures, 63 new panels, 10 updated main text panels and data from quantification of retinal neurons across 28 retinas.

Our point-by-point answers (black) to the reviewer comments (blue) are listed below and our changes to the text are indicated by yellow highlighted text in the manuscript.

Reviewer #1 (Remarks to the Author):

This manuscript from Barrie et al. is a very nicely done characterization of ipRGCs that are labeled in the GlyT2 Cre line. The authors describe the subtypes, projection patterns, spacing and neurotransmitter content of these ipRGCs, finding that they generate an independent mosaic of ipRGCs in the dorsal retina overlaid on the whole retina ipRGC mosaic. Overall, this work will be a welcome addition to the field. My concerns are addressable and outlined below.

1. The quantifications and images of the density/coverage/etc. and the anatomy are both quantitative and visually striking.
2. The title and abstract make it sound as though the paper is going to describe how this new mosaic of ipRGC encodes ground luminance. Beyond describing the fact that this mosaic exists in the dorsal retina, this question of how it encodes luminance and how that information is used is not further pursued, which was jarring as a reader. The title and abstract should be adjusted to more accurately portray the content of the paper, which is to characterize the labeled RGCs in this mouse line. The idea that the dorsal retina gets input from the ventral visual field is not, for example, a new idea.

We agree with the reviewer; the idea that the dorsal retina gets input from the ventral visual field is not a new idea. We would like to emphasize however, central to this study is the first example of a RGC subtype that is localized to a sub-region of retina which, in itself suggests that they encode only ventral vision. Therefore, by proxy - as they are ipRGCs – they likely encode ventral luminance. We do however concede that we do not examine how they encode luminance and have changed the title to reflect this and to address the request. The new title is:

A melanopsin ganglion cell subtype forms a retinal mosaic projecting to the supraoptic nucleus

We have slightly modified the abstract from the first submission but maintain the sentence which includes the phrase "... forming a substrate for encoding ground

A melanopsin ganglion cell subtype forms a retinal mosaic projecting to the supraoptic nucleus

luminance”, which we believe highlights the possibility these ipRGCs encode ground luminance, based on their location.

3. The authors state that this population primarily consists of M1 ipRGCs, but 16 of the 27 filled RGCs are ON stratifying, i.e., not M1 ipRGCs. The authors do not provide other quantification to support that this is majority M1, which they could do with their retinal density measurements since they are able to identify M1 ipRGCs using immunolabeling.

Thank you for raising this issue which relates to the n=11 OFF stratifying cells, n=11 ON stratifying cells, and n = 5 small ON stratifying cells that were analyzed for their dendritic morphology in Fig 2. The confusion arises from the sentence “These experiments revealed they are predominantly comprised of an OFF stratifying type, a structural feature of M1 ipRGCs”. This sentence is somewhat misplaced given that our numbers taken from Neurobiotin fills are not representative of the abundance or relative percentage of cells in the retina but reflect the bias towards encountering M1 or M2 ipRGCs in our Neurobiotin cell fill experiments. Small ON cells were rarely encountered in these experiments and we decided to include them to demonstrate that this line does not label purely M1 and M2 morphological variants.

Subsequent quantification from whole retinal maps has provided more concrete numbers reflecting that the number of OFF-stratifying (now referred to as sublamina-a stratifying (M1 ipRGCs)) and ON-stratifying now referred to as non-M1 ipRGCs are roughly equal (Fig 4h, Fig. S5, S6). However, M2, M4, M5, and M6 ipRGC subtypes all share sublamina-b stratification and our whole retina mapping experiments do not distinguish between these subtypes for technical reasons. To address the above confusion, we have removed the text “These experiments revealed they are predominantly comprised of sublamina-a stratifying type, a structural feature of M1 ipRGCs”. This has been replaced with “Two primary morphological types (M1 and M2) were most commonly encountered in Neurobiotin fills however, we also encountered some smaller ON-stratifying ipRGCs that resembled M5 or possibly small M4 cells (n = 3 of 25 fills).” Fig. 2 has also been modified to include *OPN4^{Cre};Brn3b^{zDTA}* M1 ipRGCs, traced with melanopsin immuno-staining to address point 5 below. No quantifiable difference exists between M1 ipRGCs in the *OPN4^{Cre};Brn3b^{zDTA};Ai9* and M1 ipRGCs identified in the *GlyT2^{Cre};Ai9*, providing further evidence that sublamina-a stratifying ipRGCs labeled in the *GlyT2^{Cre}* mouse line are indeed M1 ipRGCs.

The overall proportion of GlyT2 ipRGCs and SON-ipRGCs that are M1 vs. not should be carefully quantified.

We have carefully quantified the number of GlyT2Cre ipRGCs that are M1 vs not (sublamina-b cells) in the *GlyT2^{Cre};Ai140* animals across the entire retina (whole retina maps n = 5 retina). These results are described in Figure 4 & Fig. S5, where Fig. 4d illustrates the total number of M1 ipRGCs, Fig. 4e represents the total number of EGFP-positive SON-ipRGCs, Fig 4g represents EGFP-positive sublamina-b stratifying ipRGCs and Fig. S5 represents the whole retina maps. We also quantified the number of GlyT2-ipRGCs that are M1 vs not in animals that received AAV eye injections coding for Cre-dependent fluorescent protein (Fig. S6; n = 6 retina), revealing that the total number,

A melanopsin ganglion cell subtype forms a retinal mosaic projecting to the supraoptic nucleus

percentage, and location of M1 and non-M1 ipRGCs in the *GlyT2^{Cre}* mouse are equivalent with the numbers described in the *GlyT2^{Cre};A140* line (Fig. S5).

Additionally we quantified the number of GlyT2-ipRGCs that are M1 vs not in the whole retinas retro-labeled by SON brain injections (Fig. 6 and Fig. S12). The results are addressed in response to the comment below.

The SON ipRGCs were only loosely quantified as M1 (~97% but unclear where that number came from).

We listed the number of cells identified in line 226 at the end of the same sentence “the majority of ipRGCs labeled with these injections were OFF-stratifying M1 ipRGCs (~97%)...(n = 131 ± 16.4 M1 ipRGCs, n = 4 ± 1 non-M1 ipRGCs, n = 3 animals)”. That number came from 4 ± 1 of 131 ± 16.4 cells accounting for about 3% non-M1 ipRGCs.

It would be helpful to show confocal images supporting that the authors can truly identify stratification of SON-projecting cells (I don't doubt that they can, but they should demonstrate it).

We have provided confocal data that illustrates the OFF stratification of M1 ipRGCs identified in confocal images of melanopsin antibody staining (Figure 4b,c) and from confocal images of virus-labeled SON-ipRGCs illustrating their dendrites colored according to the layer of the retina (Fig. 7 e,f in our original submission). In the latter case, the SON-ipRGC soma, axons, and primary dendrites begin in sublamina-b with dendrites diving and branching in sublamina-a. We have now included an additional panel (Fig. 6c) to replace the previous Fig 7e,f with a side-projection illustrating the dendrites branching in sublamina-b. To address Reviewer 1, point 5 we compared the morphology of SON-ipRGCs to M1 ipRGCs labeled in the dorsal *OPN4^{Cre};Brn3b^{ZDTA}* retina (Figure 2). This provides additional evidence that SON-ipRGCs are indeed M1 ipRGCs.

Moreover, the small ON cells could be temporal M4 cells, have the authors looked at SMI32 labeling?

We have now stained 3 *GlyT2^{Cre};Ai9* retinas with the SMI-32 antibody (in addition to melanopsin antibody staining) and performed whole retina confocal microscopy mapping (Fig. S3). ~4% of the Glyt2Cre ipRGCs were SMI-32 positive suggesting that some of the small ON cells might well be temporal M4 cells but do not make up the majority of non-M1 ipRGCs identified in the *GlyT2^{Cre}* mouse.

4. The authors acknowledge that some non-dorsal ipRGCs are labeled in their retrograde injections. They give good reasons that this is the case, but still the conclusion that dorsal ipRGCs are the sole input to the SON has not been conclusively demonstrated until spillover has been ruled out. The authors should soften this language in line 237, it does not detract from the findings to do so.

A melanopsin ganglion cell subtype forms a retinal mosaic projecting to the supraoptic nucleus

We agree with the reviewer and did originally use the word “suggest” in the below sentence but we have further softened the language by including the word “may” in “Together these data suggest that the dorsal GlyT2Cre ipRGCs [may] represent the sole retinorecipient projection to the SON...”. We also changed the wording at the end of the paragraph to “Because these ipRGCs ~~represent the exclusive projection to~~ are selectively labeled from the SON, we now refer to them as SON-ipRGCs.

5. The authors briefly characterize the morphological features of SON M1 ipRGCs, but it would be useful to compare this to other dorsal M1 ipRGCs to determine if there are distinguishing morphological features of this population, this would be an important addition to our emerging understanding of M1 variability/diversity.

We agree with the reviewer that this is an interesting question but it is challenging to address carefully and beyond our current bandwidth. There is no straight-forward way to target non-SON M1 ipRGCs in *GlyT2^{Cre}* mice. We have taken the following approach, which does not completely address the reviewers request but does get us part way and somewhat addresses an earlier question about determining their M1 morphological type.

We compared the morphology of SON-ipRGCs to dorsal M1 ipRGCs labeled in *OPN4^{Cre};Brn3b^{zDTA}* mice (Fig. 2b), which contains a relatively pure population of M1 ipRGCs. While this cannot differentiate dorsal M1 ipRGCs that do not express *Cre* from the SON-ipRGCs, it does provide further evidence that SON-ipRGCs are indeed M1 ipRGCs as there is no significant difference in the parameters of their morphology that we quantified.

6. The authors do not mention whether they saw SON-ipRGC collaterals in the shell of the OPN or other targets. They should note whether each of the targets that receive GlyT2 ipRGC axons receive SON-ipRGC collaterals, this will be important in identifying 1. Potential projections of the non-M1 GlyT2 ipRGCs and 2. Give a more comprehensive overview of the innervation patterns and types of GlyT2 ipRGC inputs to each area.

We thank the reviewer for this suggestion. Our re-analysis now provides a compelling further understanding of their innervation patterns.

We have carefully quantified the central projections in *GlyT2^{Cre}* mice, including a new set of mice that received unilateral stereotactic eye injections of *AAV2-FLEX-EGFP*. Although GlyT2 ipRGCs sparsely innervate a number of distinct brain regions described in the supplementary figures, including the OPN shell (Fig. S10a-c), the zona incerta of the hypothalamus (Fig. S8b) and the stratum opticum (SO) of the superior colliculus (SC; Fig. S10d), the predominant innervation of the *GlyT2^{Cre}* ipRGCs is the outer core of the SCN, the SON, IGL and the parvocellular layer of the vLGN. The quantification that supports these conclusions is now described in Fig. 5 and Fig. S10 using normalized fluorescence values and cross-sectional fluorescence profiles. Fig. S10 specifically addressed the sparse projections to the OPN.

A melanopsin ganglion cell subtype forms a retinal mosaic projecting to the supraoptic nucleus

To address part 2 of the reviewer comment above we also performed additional quantification on the axon collaterals from GlyT2Cre and OPN4Cre mice that received central injections of *AAV_{RG}-FLEX-tdTomato* into the SON. These results are now described in Fig. S14, S15, and S16 and suggest that the IGL, SON and outer core of the SCN are co-innervated by SON-ipRGCs. As requested, the OPN was also examined. 30% of the animals showed no projections to the OPN. The remaining injections showed only sparse fibers localized to the ventral OPN shell in both the *GlyT2^{Cre}* and *OPN4^{Cre}* animals (Fig. S16) consistent with the minimal innervation pattern quantified in the anterograde labeling (Fig. S10). The sparse labelling of SON-ipRGCs and the labelling of many central neurons in *OPN4^{Cre+/-}* mice made analysis in this area particularly challenging (Fig. S16a,b).

Finally, to give a more comprehensive overview of the innervation patterns between M1 and non-M1 *GlyT2^{Cre}* ipRGCs we also compared the anterograde (M1 and non-M1 *GlyT2^{Cre}* ipRGCs) with the retrograde (M1 *GlyT2^{Cre}* ipRGC collaterals) projection patterns. These results are now illustrated in Fig. S14-16 and suggest that M1 *GlyT2^{Cre}* ipRGCs primarily innervate the IGL, SON and outer core of the SCN. The other non-M1 ipRGCs labeled in the *GlyT2^{Cre}* likely project to the IGL, parvocellular layer of the vLGN, and the localized ventral region of the dLGN. Projections to the ventral portion of the OPN shell are too sparse to interpret but quantified none-the-less (Fig. 16).

As a summary of these results, we show evidence that SON-ipRGCs primarily branch in the SCN, SON and IGL. Non M1-ipRGCs primarily innervate the IGL, vLGN, dLGN, and the OPN. We cannot however, rule out non-M1 ipRGC projections to the hypothalamus and SON-ipRGC projections to the OPN.

7. It would be useful to know, since the authors address neurotransmitter content of these SON-ipRGCs, whether they actually express GlyT2 mRNA or protein. This will provide important insights into the biology of these cells (i.e. are capable of packaging glycine) and define whether this labeling is simply a feature of the BAC line, as they suggest, but does not represent actual gene expression. This is particularly useful if others may want to use this as a marker of this ipRGC population in future studies.

We completely agree with the reviewer that it will be useful to know if SON-ipRGCs express GlyT2 mRNA or protein and we have attempted to label these ipRGCs in unison with GlyT2 protein using two separate antibodies, both of which failed to work in the retina, as shown previously (Eulenburg et al 2018).

The next potential step to addressing these questions is the subject of a follow up study. While useful, knowing the expression/lack of expression of GlyT2 mRNA or protein will not detract from the main conclusions of our study. Indeed, if GlyT2 mRNA is demonstrated, it would not immediately be clear that these ipRGCs actually express sufficient GlyT2 protein. Further examination of these questions beyond our current bandwidth for this resubmission.

Reviewer #2 (Remarks to the Author):

A melanopsin ganglion cell subtype forms a retinal mosaic projecting to the supraoptic nucleus

In this interesting and thorough study, the authors have identified a new subtype of ipRGC. By exploring the expression pattern of the retina of the GlyTr-Cre BAC transgenic mouse, the authors found that in addition to labeling inhibitory interneurons, a subset of dorsally located retinal ganglion cells were also labeled (RGCs are identified by the presence of an axon). They show that some of these RGCs as ipRGCs based on 1) co-staining with melanopsin 2) their projection targets and 3) their strong intrinsic light response. Intracellular fills show that they are a mixed population – some with dendrites stratifying in the ON sublamina and some stratifying in the OFF sublamina, the latter property consistent with them being M1-ipRGCs.

The authors then focus the rest of the study in these dorsally located OFF-stratifying M1 ipRGCs, focusing on whether they form a unique subclass from the other M1 ipRGCs with which they are intermixed. One piece of evidence is their “mosaic tiling” (more on this below). They also show they have central projection targets distinct from other M1s - namely the “outer core” of the SCN (between the shell and the central core) and that they also innervate the SON. Finally, in a set of heroic experiments, they express channelrhodopsin in all of the GlyTr-Cre RGCs and show they form glutamatergic synapses with targets in the IGL and SCN. The authors also propose a unique function for this cell type – its dorsal location indicating they are uniquely tracking luminance on the ground, though the physiological significance of this is speculative.

I have few criticisms of the technical aspects of the project – it is all extremely well done. This work represents a significant contribution to our understanding of the retina and its influence on physiology. One general comment is that the authors have used a “kitchen sink” approach for each of the submethods – which sometimes crowds out the experiments that are more definitive and leaves the reader more confused than necessary. Hence the authors are encouraged to streamline when possible.

Below are some specific comments regarding aspects of the study that need to be clarified.

1. The authors emphasize that the presence of “a regularly spaced mosaic” (e.g. line 88, line 243) and argue this organization is indicative of correct identification of a cell type. This mosaic argument is based on the presence of an exclusion zone in the density recovery profile - DRP (e.g. SON-ipRGCs form an ordered mosaic on the dorsal retina). The authors are strongly encouraged to read recent review Keeley 2020 (and Reese 2015) that make the case that the presence of an exclusion zone is not conclusive evidence that a retinal mosaic is regular and significantly different from random, especially with sparsely distributed cells such as SON-ipRGCs.

This is our mistake for focusing on the regularity of the mosaic as evidence for a subtype, when in actuality we meant to emphasize that the combined molecular, morphological and territorial spacing of the dendrites argues for a subtype. We thank the reviewer for raising this important point and have read the above references and agree that the presence of an exclusion zone does not alone indicate regularity in a

A melanopsin ganglion cell subtype forms a retinal mosaic projecting to the supraoptic nucleus

mosaic. Our analysis shows that despite their luminance-detecting function, ipRGCs likely exhibit territorial dendritic spacing and uniform coverage, which is surprising for a non-image forming cell type. To address this and to address the below point, we have removed statements about the regularity of their soma spacing. We also modified the following sentence in the text (~ line 298):

“*GlyT2^{Cre}* and *OPN4^{Cre}* mice exhibited a clearly defined exclusion zone around the soma, which indicates their cell bodies **may** be regularly spaced (Fig. 7 e,f; Fig. S13 a,b), **although the presence of an exclusion zone is not conclusive evidence that a retinal mosaic is regular.**”

One alternative is to compare the measured DRP with the DRP computed from a random distribution of cells matched for soma size and density. That said, I don't think the lack of “regular” mosaic will significantly weaken their results – but the authors are cautioned on relying too much on this method for defining a cell type and should remove reference to the soma organization being “regular” throughout. (I would have classified this as a “minor” point except that the authors have emphasized this analysis throughout.)

Thank you for the suggestion. We have now compared the measured DRP with a random distribution of cells and our results are contained in Figure 7. Here we simulated a random distribution of cells matched for density and soma size. The density recovery profiles for the random modeled distribution exhibited high-density values with proximal soma distances (< 100 μm) similar to the unspecified M1 ipRGCs and non-SON projecting M1 ipRGCs (Fig. 7e,f). This is unlike the SON-ipRGCs labeled with retro-injections in *GlyT2^{Cre}* and *OPN4^{Cre}* mice, which exhibited a clearly defined exclusion zone around the soma.

We also calculated a regularity index using nearest-neighbor measurements. Our random modeled distribution had a ratio consistent with a theoretical random distribution (NN regularity indexes < 1.91) whereas SON-labeled retina had a ratio of 3.05 ± 0.27 consistent with other retinal cell types with mosaic distributions (Reese et al. 2015, Keeley et al. 2020).

M1 ipRGCs have sparse and asymmetric dendrites making receptive field extrapolations and mapping challenging. These experiments regarding soma distribution are not interpreted as proof of regularity or mosaicism, they used to show that SON ipRGCs exhibit distribution characteristics shared by tiling retinal subtypes. The language in the text has been softened appropriately (as described above in response to point 1) whereas suggested we have removed text referring to regularity and focused instead on territorial tiling.

2. As is clearly the case here, BAC-transgenic in general cannot be used to cleanly identify cell types. Indeed the multiple other types of cells labeled in this line – in particular the non-M1 ipRGCs, can complicate the interpretation of several experiments. The authors need to quantify what percentage of labeled RGCs in the *GlyT2-cre* mouse

A melanopsin ganglion cell subtype forms a retinal mosaic projecting to the supraoptic nucleus

line are non-ipRGCs (line 116). For example, in Figure 1, they do not give the percentage of RGCs expressing eGFP (or td tomato in Figure S1) that are melanopsin positive nor do they give a quantification of percent of cells they patched that have a light response.

We agree with the reviewer. In all our experiments, we have found little evidence that there are any non-ipRGC ganglion cells in this line however we did not clearly present this in our original submission. To address this request, we co-stained GlyT2Cre;Ai9 retina with RBPMS, a specific marker for retinal ganglion cells. We find that 99.4% of the RBPMS-positive cells labeled in the GlyT2Cre;Ai9 mouse express melanopsin, suggesting that they are exclusively ipRGCs. This result is now reported in Fig. 3i-k with whole retina maps provided in Fig. S2. Furthermore, 9 cells were targeted for single cell patch, all were intrinsically photosensitive in the presence of synaptic blockers.

3. As noted by the authors, a subpopulation of dorsal ipRGCs were previously described as expressing PACAP (Hannibal et al 2002) (line 478). This point feels rather buried in the discussion when there have been several studies regarding the role of PACAP in the SON and its influence on other circadian functions. Rather than a side point, this is important context for the current study and should be brought up in the intro. Perhaps more importantly, it seems straightforward for the authors to test whether these cells are the same subpopulation - namely, are the SON-ipRGCs PACAP positive? Knowing whether these cells are PACAP positive would also provide a potential way to genetically identify these ipRGCs, as so far there have been no specific marker genes identified for SON-ipRGCs.

Unfortunately we cannot label our ipRGCs with a commercially available PACAP antibody, which we tested with no success. The PACAP antibody in the Hannibal et al. 2002 manuscript is no longer available and although their result is compelling, as their distribution is strikingly similar to the distribution we find in the SON-ipRGCs, this might also be because ipRGCs are more numerous in the superior retina in general and sub-optimal staining of all ipRGCs might appear similar.

We approached the reviewer's question by performing eye injections in the *PACAP-2A^{Cre}* mouse with *Cre*-dependent *AAV-ChR2-EYFP*. We used virus because while this mouse is a knock-in and not a BAC-transgenic, crossing to a *Cre*-dependent reporter line results in most RGCs being labeled, suggesting that they express PACAP transiently or at low levels. In addition, most central neurons were also labeled in the *PACAP-2A^{Cre}*;reporter crosses. Our results however, suggest that the identity of PACAP-expressing ipRGCs in the mouse retina may differ from that reported in Hannibal et al. 2002. Most RGCs labeled in this way were melanopsin-negative (Supplemental Fig 4), although we did see some weakly immunopositive ipRGCs with large somas (presumptive M4 ipRGCs) labeled. The central innervation of these RGCs in the LGN confirms these results with most axon terminals appearing in the dorsal LGN, with some terminals in the ventral LGN, but no terminals in the intergeniculate leaflet. Together these results suggest that if *PACAP-2A^{Cre}* line faithfully represents the complement of PACAP-containing RGCs in the retina,

A melanopsin ganglion cell subtype forms a retinal mosaic projecting to the supraoptic nucleus

they do not overlap with the ipRGCs labeled in *GlyT2^{Cre}* mice. We have changed the text and the discussion to reflect this modified finding.

Line 169 – “Dorsal *GlyT2^{Cre}* ipRGCs did not overlap with RGCs labeled in mice where *Cre* recombinase is driven by the peptide pituitary adenylate cyclase-activating polypeptide (PACAP; Fig. S4), which are abundant in the dorsal rat retina⁴².”

Line 488 – “The peptide PACAP is present in a dorsal population of ipRGCs in the rat⁴² and both PACAP and the PACAP receptor PAC₁ are expressed in the SON^{77,78}. Additionally, PACAP positive retinal hypothalamic tract terminals are localized to a SCN region that resembles the outer-core projections of SON ipRGCs. However our own experiments to determine if the SON-ipRGCs and other *GlyT2^{Cre}* expressing ipRGCs overlap with PACAP-expressing RGCs (Fig S4) failed to identify overlap, suggesting that PACAP-expression may be different in the mouse retina.”

4. Similarly, the analysis of the various projection targets is quite detailed and impressive but seems critical to know if non-ipRGCs or ON-stratifying ipRGCs comprise some of these targets, particularly in areas outside of exclusive ipRGC target regions such as zona incerta (e.g supplementary Figure 5).

Reviewer 1 also expressed similar interest which is addressed in detail above (Reviewer 1, response 6). To strengthen the conclusions regarding central sites of ipRGC projection we first confirmed that the RGCs labeled in the *GlyT2^{Cre}* mouse are exclusively ipRGCs using RBPMs and melanopsin immunostaining (Fig. 3i-k) (Addressed above: Reviewer 1, point 6; Reviewer 2, point 2). We then performed stereotactic unilateral *Cre*-dependent AAV eye injections in *GlyT2^{Cre}* mice and quantified the sites of innervation using normalized fluorescence values and cross-sectional fluorescence profiles (Fig. 5 & Fig. S10). We quantified the retinas of these animals to determine the percentage of M1 and non-M1 ipRGCs labeled (Fig. S6). We then performed similar analysis in SON-injected *GlyT2^{Cre}* and *OPN4^{Cre}* animals, determining the percentage of M1 and non-M1 ipRGCs labeled in the retina (Fig. 6 & Fig. S12) and quantifying the collateral projections in the brain in comparison to anterograde projections obtained in the eye injected animals (Fig. S14,S15,S16).

To summarize these results we find that the RGC population in the *GlyT2^{Cre}* mouse line are exclusively ipRGCs that comprise equal numbers of M1 and non-M1 ipRGCs. The predominant innervation of the *GlyT2^{Cre}* ipRGCs is the outer core of the SCN, the SON, IGL and the parvocellular region of the vLGN, exhibiting the highest normalized fluorescence values. Projections to the zona incerta of the hypothalamus (Fig. S8b), a localized subregion the medial dLGN, the ventral portion of the OPN shell (Fig. S10a-c) and the lateral portion of the stratum opticum (SO) of the superior colliculus (SC) (Fig. S10d) were also observed, but represent only minor projections and were therefore presented in the supplements. SON-targeted brain injections retro-labeled dorsal M1 ipRGCs in both the *GlyT2^{Cre}* and *OPN4^{Cre}* mouse lines (SON ipRGCs). SON-ipRGCs have collateral innervation to the outer core of the SCN and a quantifiable projection to the IGL. Comparison of these projection patterns with eye-injected mice suggest that non-M1 ipRGCs project primarily to the parvocellular region of the vLGN, the IGL, the

A melanopsin ganglion cell subtype forms a retinal mosaic projecting to the supraoptic nucleus

ventral subregion of the dLGN, and the OPN. Our results also revealed that the projections to the SCN from one eye are bilateral but the projections to the SON are unilateral.

5. Figure 8 and the supporting text is a bit hard to figure out. First, the figure caption doesn't match the lettering on the figure. Second, line 335 mentions mapping locations of postsynaptic SON neurons by matching to a DIC image but this is not in the figure.

We have changed Fig 8 and the figure legend to address this including making sure that the lettering and the caption mislabeling was corrected. Second, we have added a Supplementary figure (Fig. S17) with the DIC images used to demonstrate the SCN and SON neuron locations.

It is also not clear what is measured in a "cell attached voltage clamp mode" – this is just an unclamped AP, correct? The value of these measurements is not clear particularly when there are whole cell voltage clamp experiments as well.

The reviewer is correct, the cell attached voltage clamp recordings only show APs which are evoked with a short delay by ChR2 activation. We included these recordings to demonstrate that ChR2 activation is sufficient to evoke post-synaptic action potential generation and this is blocked by excitatory antagonists, strongly suggesting a primarily glutamatergic drive from ChR2-expressing axons. This is prior to breaking into the cell, which provides additional information illustrating this synaptic relationship is present without intracellular dialysis, and blockade of action potentials with QX-314.

We added the following text to highlight these are unclamped and reduce any confusion that we might be claiming the APs are voltage clamped.

(Line 362). "Prior to break-in, in cell-attached voltage clamp mode photo-stimulation activated robust extracellular unclamped action potential (AP) currents, which demonstrates that the SCN neuron was depolarized beyond its action-potential threshold by the release of an excitatory transmitter."

Also the motivation for using the *Opn4-Cre* in Part O is not clear for these purposes. Finally, no quantification is given for Parts N and O so the conclusions for these coronal sections seem rather limited and the authors should consider removing.

Our motivation for using *OPN4Cre* was due to low numbers of *GlyT2^{Cre};Ai32* mice and an attempt to increase our numbers of SON recordings which have a low hit rate. We now include an additional 6 recordings from 6 mice and illustrate pooled data from these recordings. We chose to keep these recordings in the manuscript as they demonstrate the first example of targeted ChR2 activation of ipRGC synaptic release in the SON and this area is a central theme of our findings.

6. Why are the authors including Figure 9a-g? This seems like a poor experiment since they ultimately decide that there are multiple sources of input to the IGL neurons and therefore not that interpretable regarding retinal inputs. I would recommend eliminating

A melanopsin ganglion cell subtype forms a retinal mosaic projecting to the supraoptic nucleus

this part of the figure and stick to the more interpretable component regarding Figure 9h-n. If there is some other motivation for 9a-g, then the authors need to provide it. And if 9a-g is retained, the summary data in d-e should make clear which data was measured in the presence of single synaptic blockers vs combinations.

Fig. 9ag were our original experiments and subsequent eye injections were performed to rule out other central neurons projecting to the IGL, illustrating the progression of our experiments. We have moved 9a-g to the supplements (Fig. S17) and the figure focuses on the eye injection experiments alone.

7. Supplementary figure 7d - remove the brown points indicating “predictive SON projecting,” since that unreasonably assumes a priori that the mosaic spacing of SON-ipRGCs in the shown area is regular.

We have removed the “predictive SON projecting” as suggested.

8. Lines 282/supplementary Figure 8 – it was unclear the motivation behind looking at these projections or exactly what is being shown here.

Thank you for pointing out the lack of clarity here. Our goal was to illustrate the collateral projections from SON-ipRGCs that are labeled from SON injections. Given that the retina contains predominantly sublamina-a stratifying SON-ipRGCs, and consist of all the M1 ipRGCs labeled in the *GyT2^{Cre}* mouse line, we reasoned examining other brain regions in these SON-injected animals might provide insight into the central projection locations of SON-ipRGCs without the ON stratifying cells (as requested by reviewer 1, and 2). This aspect has now been expanded to include careful quantification of collateral brain projections in the SON-injected *GlyT2Cre* and *OPN4Cre* mice. Furthermore, central projections of collateral innervation are compared with the patterns quantified from anterograde eye injections. The results of these experiments are described in Fig. S14-S16, and above in response to review 1, comment 6 and review 2 comment 4.

9. Minor point: In Figure 2, an analysis of dendrites is used to make the case that OFF-stratifying labeled cells are similar morphology to M1s (line 133) – however no reference is given for what is known about M1 dendritic morphology and how strong this mapping is. The authors may at least want to add the reference or state explicitly that the morphology is just similar at a gross assessment.

Thank you for pointing this out. In addition to the additional quantification of dendritic stratification, and additional analysis of *OPN4^{Cre};Brn3b^{zDTA}* retinas that were used to address Reviewer 1 comment 5, we have added the following references that identify M1 ipRGCs as OFF stratifying (line 130):

A melanopsin ganglion cell subtype forms a retinal mosaic projecting to the supraoptic nucleus

Reference 34: Schmidt, T. M. & Kofuji, P. Functional and morphological differences among intrinsically photosensitive retinal ganglion cells. *J Neurosci* **29**, 476-482, doi:10.1523/jneurosci.4117-08.2009 (2009).

Lee, S. K., Sonoda, T. & Schmidt, T. M. M1 Intrinsically Photosensitive Retinal Ganglion Cells Integrate Rod and Melanopsin Inputs to Signal in Low Light. *Cell Rep* **29**, 3349-3355.e3342, doi:10.1016/j.celrep.2019.11.024 (2019).

Wu, X. S. *et al.* Morphological alterations of intrinsically photosensitive retinal ganglion cells after ablation of mouse photoreceptors with selective photocoagulation. *Exp Eye Res* **188**, 107812, doi:10.1016/j.exer.2019.107812 (2019).

Reviewer #3 (Remarks to the Author):

In this manuscript, the authors report a subtype of ipRGCs that form a regularly spaced mosaic limited to a dorsal region of the retina, selectively projecting to the supraoptic nucleus. Using a combination of confocal microscopy, anterograde and retrograde labelling, patch clamp recordings and optogenetics, the authors show that ipRGCs can be further subdivided and comprise independent subtypes that tile retinal space, like other conventional RGCs reported previously, and suggest that mice devote additional melanopsin-dependent processing power to their ventral visual field. I find this manuscript provides interesting findings and a perspective on a possible role of the ipRGCs and their specific projections to the lateral hypothalamic brain regions. I only have minor comments, which largely are requests for additional analysis details and further clarifications of their findings for better presentation.

- Lines 100-121 (Fig1 and S1) I wonder whether all the analysis regarding the projections of RGC axon terminals in the brain were done manually and visually. Conventional manual processing is highly subjective and sample-specific, and even precise comparison across sample animals is impossible. Thus, can you show any quantitative data or analysis to confirm your conclusion of systematic mapping? – If possible, it will be helpful to show some numbers and statistics such as the number of cells measured each RGC and the average portion of the projection in the target area, etc. in each animal. Another possible scenario is to use some pre-published algorithms and software (For example, see Song and Choi *et al.* *Cell Reports* 2020) to compare datasets from different mice successfully on a properly calibrated reference space. In any cases, it would be good to provide detailed numbers and statistics so that other researchers can reproduce your findings and compare with.

Thank you for rising this important point and we agree. To improve the repeatability and provide quantifiable and statistical descriptions of the ipRGCs in this study we performed unilateral stereotactic eye injections in an additional cohort of GlyT2Cre animals. These retinae were carefully quantified to determine the distribution, count and ratio of M1 and non-M1 ipRGCs in whole retina maps (Fig. S6). The projection sites

A melanopsin ganglion cell subtype forms a retinal mosaic projecting to the supraoptic nucleus

were quantified (Fig. 5; Fig. S10). Quantification was performed by calculating normalized average fluorescence values for pertinent brain regions which were localized using DAPI staining, immuno-staining of arginine vasopressin, and a Mouse Brain Coordinate Atlas. Additionally, cross-sectional fluorescence through brain structures were also used to compare patterns of innervation within relevant brain regions/nuclei and were compared to the labeling pattern of all RGCs generated by CTB eye injection (Fig.5 and Fig. S10). These approaches strengthen the conclusions in the paper by providing objective and quantifiable descriptions of the GlyT2Cre ipRGC innervation pattern in the brain. For more detail see Reviewer 2 response 4.

Lines 135-145 (Fig 3) This is one of the main results of this study, but it seems that only a single animal case is shown. How many different animals were used to confirm this result, and how the result varies across animals? Particularly, on the right graph of Fig 3d-f (density at each retinal location along the dorso-ventral axis), you can show the result in each animal separately and then plot the averaged one.

Additional quantification has also been expanded into Fig 4. Fig. 4d-g contains a single representative retina map and the average of n=5 retinas in the dorso-ventral axis graphs (right of each map). The individual maps of the additional retina are now provided in Fig. S5. Figure 4i demonstrates the above point for M1 ipRGCs from n = 5 animals. Figure 4k contains additional analysis.

- Lines 174-187 (Fig 5) The results in Fig 5 are mostly described qualitative using images only, which requires a subjective visual inspection. Probably, the bottom line of the result that authors claimed might be acceptable as it is, but it would be very helpful to try some quantitative analysis or plots of graphs that describes everything in more precise form. In addition, for details, you can describe how did you define each brain region, count the number of projections in each area, and how robust your observation across different animals, etc.

Thank you for the suggestion. Much of this has now been addressed as described above (Reviewer 1 point 6; Reviewer 2 point 4, Reviewer 3, point 1). Brain regions were quantified using normalized average fluorescence values of brain regions. Regions were localized using DAPI staining, immuno-staining of arginine vasopressin, and a Mouse Brain Coordinate Atlas. Although there was some variability in the level of fluorescence the pattern of innervation was consistent across all animals. Statistical comparisons were made between contralateral and ipsilateral sites. Furthermore, projection patterns from the anterograde eye injections were statistically compared with the collateral pattern of innervation achieved through SON brain injections in the *GlyT2^{Cre}* and *OPN4^{Cre}* mice. The results and methods have been expanded to reflect this new analysis.

Lines 253-266 (Fig 7 and S7) Here it seems important to describe the mosaics and

A melanopsin ganglion cell subtype forms a retinal mosaic projecting to the supraoptic nucleus

corresponding analysis together, so why don't you include materials in Supplementary Figure 7d in the main figure? Instead of the current panels Fig. 7a and b that are too small to see some details of mosaics structure, probably you can show an illustration of cell mosaics as that of the RGC receptive fields previously reported (For example, see Fig 1. in Anishchenko et al. Journal of Neurophysiology 2010), using the cell density and the coverage factor estimated here. This information is important enough to be in the main manuscript, and may help readers to more easily examine a spatial organization of the population.

We agree that the results previously in Fig. S7 (now S13) are important and we attempted to incorporate these images into the main text as suggested. However, in the end we felt that when combined with the quantification provided in Fig. 7e-i, as well as the zoomed in example images in 7c, d, including the large panel from (now) Fig. S12 overcrowd the figure and so we think this works better as a supplement.

To aid in interpretation of the tiling structure we recreated Fig. 7a,b to increase the contrast of the mosaics to be similar to the figure in Anischenko Fig.1 but chose to retain the underlying retina confocal images.

- Lines 402-404

“when SON-ipRGCs are subtracted from retinal density maps, the dorso-ventral density gradient disappears and the number of M1 ipRGCs is equivalent in both hemispheres.”
: This seems quite an interesting observation, but I can't find a relevant plot for this.

These results are presented in Fig. 4i,j with an additional Fig. 4k plotting the distribution of average densities along the dorsal-ventral axis with and without EGFP+ M1 ipRGCs. Individual retina maps are presented in Fig. S5 with c illustrating that an equal number of dorsal and ventral M1 ipRGCs remain when *GlyT2^{Cre}* ipRGCs are removed. Finally, graphs depicting the mean of differences comparing number of cells (dorsal versus ventral) are presented in Fig. S5 d,e.

Minors

Lines 170-178.

Figures need to be rearranged. Fig S5 and S6 appear before S3 in the main text.

These figures are now rearranged, thank you for picking up on this.

Fig 3. Some panels are missing or mislabeled. Panels h and i should be g and h.
Fig. 8. Panels 8c and d are too small to get any information visually. Can you add an inset for a “zoomed-in” local shot for better visibility?

Fig. 3 has been corrected. Fig. 8d represents a zoomed in image of fig. 8c. We have included a white box to emphasize this.

REVIEWER COMMENTS

Reviewer #1 (Remarks to the Author):

I very much appreciate the new experiments, analyses, and clarifications in the resubmitted manuscript. The authors clearly did their best to address the reviewer concerns. I have no further comments, and congratulate the authors on this very nice work.

Reviewer #2 (Remarks to the Author):

All of my concerns (and then some) were addressed in the revision, which contains a phenomenal amount of new data and analysis. There revisions to the text as well have greatly clarified the motivation for many of the experiments. This is a very fantastic study - the authors are to be commended.

Reviewer #3 (Remarks to the Author):

I thank the authors for their comprehensive response to the reviews. The authors thoroughly addressed all of the concerns raised in the initial version of manuscript. I have no further concern on this manuscript.

A melanopsin ganglion cell subtype forms a dorsal retinal mosaic projecting to the supraoptic nucleus

Following revision:

REVIEWERS' COMMENTS

Reviewer #1 (Remarks to the Author):

I very much appreciate the new experiments, analyses, and clarifications in the resubmitted manuscript. The authors clearly did their best to address the reviewer concerns. I have no further comments, and congratulate the authors on this very nice work.

Reviewer #2 (Remarks to the Author):

All of my concerns (and then some) were addressed in the revision, which contains a phenomenal amount of new data and analysis. There revisions to the text as well have greatly clarified the motivation for many of the experiments. This is a very fantastic study - the authors are to be commended.

Reviewer #3 (Remarks to the Author):

I thank the authors for their comprehensive response to the reviews. The authors thoroughly addressed all of the concerns raised in the initial version of manuscript. I have no further concern on this manuscript.

Response: We thank all of the reviewers for their constructive input.